# An open-like conformation of the sigma-1 receptor reveals its ligand entry pathway

Fuhui Meng[1,2], Yang Xiao[1,2], Yujia Ji[1], Ziyi Sun [1✉] & Xiaoming Zhou [1✉]

The sigma-1 receptor ($\sigma_1$R) is a non-opioid transmembrane receptor which has been implicated in many diseases, including neurodegenerative disorders and cancer. After more than forty years of research, substantial progress has been made in understanding this unique receptor, yet the molecular mechanism of its ligand entry pathway remains uncertain. Published structures of human $\sigma_1$R reveal its homotrimeric organization of a cupin-fold $\beta$-barrel body that contains the ligand binding site, a carboxy-terminal V-shaped two-helix bundle, and a single amino-terminal transmembrane helix, while simulation studies have suggested a ligand entry pathway that is generated by conformational rearrangements of the cupin-fold domain. Here, we present multiple crystal structures, including an open-like conformation, of $\sigma_1$R from *Xenopus laevis*. Together with functional binding analysis our data suggest that access to the $\sigma_1$R ligand binding site is likely achieved by protein conformational changes that involve the carboxy-terminal two-helix bundle, rather than structural changes in the cupin-fold domain.

[1] Department of Integrated Traditional Chinese and Western Medicine, Rare Diseases Center, State Key Laboratory of Biotherapy, West China Hospital, Sichuan University, Chengdu, Sichuan 610041, China. [2]These authors contributed equally: Fuhui Meng, Yang Xiao. ✉email: ziyi.sun@scu.edu.cn; x.zhou@scu.edu.cn

The sigma-1 receptor ($\sigma_1$R) is a small, unique integral membrane receptor that is localized primarily in the endoplasmic reticulum (ER)[1–3]. $\sigma_1$R responds to a structurally diverse array of synthetic ligands such as (+)-pentazocine (agonist) and haloperidol (antagonist)[2]. It interacts with various effector proteins[1], including ion channels[4–6] and G-protein coupled receptors[1,7–9], and it is implicated in many diseases, including neurodegenerative disorders[10–12] and cancer[13,14]. Identified in 1976[15] and cloned in 1996[16], its first atomic structure was solved in 2016[17]. The structural information, together with information gathered from functional studies, has provided insight into the key mechanistic elements of $\sigma_1$R's function[2,3,18]. However, critical mechanistic details of how ligands access their binding site in $\sigma_1$R remain enigmatic.

Recent crystal structures of human $\sigma_1$R (h$\sigma_1$R) show that h$\sigma_1$R adopts a homotrimeric configuration, with each protomer comprising a single transmembrane helix ($\alpha$1) at the amino-terminus, followed by a cupin-fold[19] $\beta$-barrel body containing the ligand-binding site, and a membrane-adjacent V-shaped two-helix bundle ($\alpha$4/$\alpha$5) at its carboxy-terminus covering the cupin-fold domain like a lid[17,20] (Fig. 1a). Currently, there are mainly two pathways proposed for ligand entry in $\sigma_1$R[17,20,21]. The first proposed pathway (PATH1) would allow a ligand to enter $\sigma_1$R through structural rearrangements of the cupin-fold domain, which involves unfolding and refolding of the $\beta$ barrel; the second proposed pathway (PATH2) would allow a ligand to access its binding site in $\sigma_1$R through the opening between $\alpha$4 and $\alpha$5 (Fig. 1b).

Recent molecular dynamics simulations support PATH1, as the simulations revealed two drastic conformational changes in h$\sigma_1$R that break hydrogen bonds of multiple $\beta$ sheets of the cupin-fold domain to expose the ligand-binding site[20] (Fig. 1a). However, considering the energy expenditure required to disrupt the $\beta$ barrel, and the entropic penalty incurred when exposing the largely hydrophobic interior of the cupin-fold domain to the aqueous environment, PATH1 seems rather energetically unfavorable. The results of another computational study involving steered molecular dynamics revealed that pulling a ligand (PD144418) out of the binding pocket of h$\sigma_1$R required less initial force in PATH1[21]. However, such a simulated conformational change of $\sigma_1$R may not represent a physiological condition. Furthermore, the unfavorable energy expense would also accompany the pulling process in PATH1 when it breaks the $\beta$ barrel.

To provide experimental evidence to assess ligand entry to $\sigma_1$R we present multiple crystal structures of $\sigma_1$R from *Xenopus laevis* (xl$\sigma_1$R). Together with mutagenesis and binding analysis, the structural data support the notion that ligand entry to $\sigma_1$R is achieved through PATH2.

## Results

**The closed conformation of xl$\sigma_1$R.** To better define the ligand entry pathway in $\sigma_1$R, our initial strategy was to capture a $\sigma_1$R structure in its open conformation. To achieve this goal, we screened expression for ten $\sigma_1$R homologs from different species and crystallized well-behaved clones without adding any known

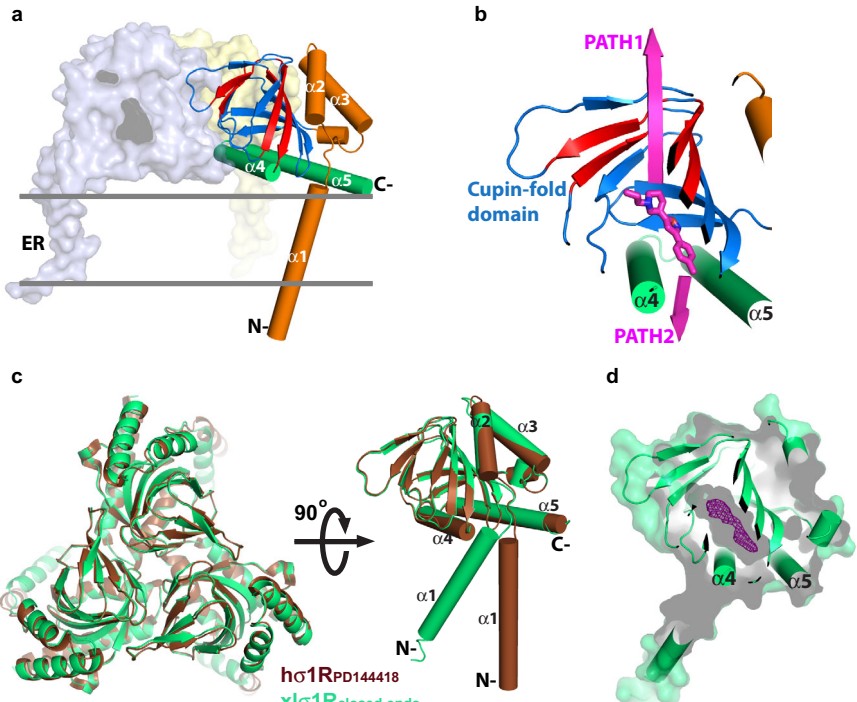

**Fig. 1 The structure of xl$\sigma_1$R$_{closed-endo}$ compared to h$\sigma_1$R$_{PD144418}$. a** The structure of the h$\sigma_1$R$_{PD144418}$ (PDB: 5HK1) homotrimer is viewed parallel to the membrane, with two protomers rendered in the surface mode and one in the cartoon. The helices are rendered as cylinders, and are labeled as $\alpha$1–$\alpha$5 from the amino- (N-) to the carboxy- (C-) terminus throughout the paper. The cupin-fold domain is rendered as a blue $\beta$ barrel, with the red sheets indicating the hydrogen bonds to disrupt for ligand entry as suggested previously[20]. The C-terminal two-helix bundle ($\alpha$4/$\alpha$5) is colored in green, while the N-terminal part is orange. The relative position of the ER membrane is indicated by two gray lines. **b** The close-up view of one protomer from panel (**a**). The bound ligand, PD144418, is displayed as magenta sticks. The two pathways proposed for ligand entry are indicated by two magenta arrows, PATH1 and PATH2. **c** Left, superposition of the xl$\sigma_1$R$_{closed-endo}$ trimer (in green) and the h$\sigma_1$R$_{PD144418}$ trimer (in brown) viewed perpendicular to the membrane from the cupin-fold side. Right, superposition of one xl$\sigma_1$R$_{closed-endo}$ protomer and one h$\sigma_1$R$_{PD144418}$ protomer viewed parallel to the membrane. **d** One xl$\sigma_1$R$_{closed-endo}$ protomer is rendered in both cartoon and surface modes in a "slab" view, in which the internal cavity is displayed as a gray shadowy compartment within the $\beta$ barrel. The purple mesh shows the simulated annealing $F_o$–$F_c$ map contoured at 3.0 $\sigma$ level corresponding to an unidentifiable molecule.

ligand for structure determination. In the end, we solved a structure for the wild-type $\sigma_1R$ from *Xenopus laevis* (xl$\sigma_1$R), which shares 67% sequence identity and 89% sequence homology with human $\sigma_1R$ (h$\sigma_1$R) (Supplementary Fig. 1), to 3.20 Å, and termed it xl$\sigma_1R_{closed-endo}$.

The xl$\sigma_1R_{closed-endo}$ structure was solved in the space group P2$_1$ with each asymmetric unit containing 12 protomers (Supplementary Fig. 2a), which deviate little from each other with all-atom RMSDs of only ~0.2 Å between the protomers. Like all reported structures for h$\sigma_1$R, xl$\sigma_1$R also crystallizes as a homotrimer in xl$\sigma_1R_{closed-endo}$, which resembles closely the h$\sigma_1$R structures, especially the ones complexed with an antagonist such as PD144418 (h$\sigma_1R_{PD144418}$, PDB 5HK1), except the orientation of transmembrane helix α1 (Fig. 1c). Superposition of individual protomers of xl$\sigma_1R_{closed-endo}$ and h$\sigma_1R_{PD144418}$ excluding α1 yielded an all-Cα RMSD of 0.35 Å, indicating a high degree of structural conservation among $\sigma_1$Rs from different species (Fig. 1c). However, xl$\sigma_1R_{closed-endo}$ failed to achieve our initial goal as it was captured in a closed conformation similarly to h$\sigma_1R_{PD144418}$ (Fig. 1d). Surprisingly, though no known ligand was added during purification or crystallization of xl$\sigma_1$R, a clear electron density within the ligand-binding pocket indicates that xl$\sigma_1R_{closed-endo}$ was bound by an unknown molecule (Fig. 1d and Supplementary Fig. 2b), which may have an endogenous origin. Unfortunately, the current resolution of xl$\sigma_1R_{closed-endo}$ (3.20 Å) provides only limited details about the shape of the electron density (Fig. 1d and Supplementary Fig. 2b), making it impossible to reveal readily the molecular identity of this unknown molecule, and therefore the electron density was not modeled in xl$\sigma_1R_{closed-endo}$.

**The crystal packing of xl$\sigma_1R_{closed-endo}$ prohibits the PATH1 formation during ligand binding**. In each asymmetric unit of xl$\sigma_1R_{closed-endo}$ there are four homotrimers of xl$\sigma_1$R, which are organized in a diamond shape with the cupin-fold domains packing against each other, whereas the transmembrane helices (α1s) mediate crystal contacts between adjacent asymmetric units (Supplementary Fig. 2a). In such a packing mode, each trimer has one protomer whose cupin-fold domain is tightly buried by two adjacent cupin-fold domains (Fig. 2a). For example, the tip region of the cupin-fold domain (Trp133–Tyr144) of Protomer C is in close contact with Protomer A (Thr106–Thr110) and Protomer D (Lys139–Tyr144) (Fig. 2b). As a result, the steric hindrance from Protomers A and D would prevent the tip region of Protomer C from moving away from the rest of its cupin-fold body, a conformational change that is required for the PATH1 formation in $\sigma_1$R as suggested in previous simulations[17,21]. Therefore, one protomer in every trimer would not be able to open in this packing mode if PATH1 is the ligand entry pathway in $\sigma_1$R. Then if a ligand was added to these packed crystals, it would not be able to access the ligand-binding site in one-third of the protomers.

To test this hypothesis experimentally, we soaked two known ligands of $\sigma_1$R, PRE084 (agonist) or S1RA (antagonist), directly into the xl$\sigma_1R_{closed-endo}$ crystals, and solved both structures, xl$\sigma_1R_{closed-PRE084}$ and xl$\sigma_1R_{closed-S1RA}$, respectively. Soaking of either ligand changed little on the xl$\sigma_1$R crystal packing. Both xl$\sigma_1R_{closed-PRE084}$ and xl$\sigma_1R_{closed-S1RA}$ are almost identical to xl$\sigma_1R_{closed-endo}$ with all-atom RMSDs < 0.25 Å when aligning the three structures (Supplementary Fig. 2c). However, in both xl$\sigma_1R_{closed-PRE084}$ and xl$\sigma_1R_{closed-S1RA}$, each of the three protomers in the trimeric assembly features a ligand (PRE084 or S1RA) bound in the ligand-binding pocket (Fig. 2c and Supplementary Fig. 2d, e). This result demonstrates that soaking of the tightly packed crystals of xl$\sigma_1R_{closed-endo}$ with the ligands led to the replacement of the unidentified molecule in each protomer,

indicating that ligand access via proposed PATH1 is highly unlikely and that the conformational changes proposed for PATH2 seem more likely to facilitate ligand entry to each individual binding site.

**An open-like conformation of xl$\sigma_1$R that is consistent with PATH2**. In our continued search for an open structure of xl$\sigma_1$R, crystals with an appearance (cube-like) different from the xl$\sigma_1R_{closed-endo}$ crystals (cuboid-like) were obtained and their structure was determined to 3.56 Å (xl$\sigma_1R_{open-endo}$). The xl$\sigma_1R_{open-endo}$ structure was solved in the P2$_1$2$_1$2$_1$ space group, with each asymmetric unit containing 12 protomers. These protomers are nearly identical to each other with all-atom RMSDs < 0.2 Å between the protomers. Similar to all $\sigma_1$R structures, xl$\sigma_1R_{open-endo}$ also assembles as homotrimers (Fig. 3a). Each asymmetric unit contains four trimers, which are packed in a tetrahedron shape through the α1–α1 contacts (Supplementary Fig. 3a).

Structural comparison between xl$\sigma_1R_{open-endo}$ and xl$\sigma_1R_{closed-endo}$ shows that the two structures are very similar, with the exception of two regions. One region consists of the transmembrane helix α1, which swings ~18° away from the trimer center in xl$\sigma_1R_{open-endo}$ compared to xl$\sigma_1R_{closed-endo}$ (Fig. 3a). Structural alignment between the two structures excluding the α1 region yielded an all-atom RMSD of 0.34 Å. This orientation difference of α1 may be due to crystal packing, as α1 in the h$\sigma_1$R structures exhibits different orientations even between different protomers of the same structure[17,20], e.g., h$\sigma_1R_{PD144418}$ (Fig. 3b). The other major difference lies in the carboxy-terminal two-helix bundle, α4/α5. In xl$\sigma_1R_{open-endo}$, the α4 helix rotates slightly away from α5, and the side chain of Tyr203 from α5 adopts a *gauche+* χ$_1$ rotamer to point to the cupin-fold domain (Fig. 3c), thus creating an opening between α4 and α5 that generates a direct pathway for the ligand from the outside milieu to the ligand-binding site (Fig. 3d). Such an access pathway is reminiscent of that proposed in PATH2 (Fig. 1b). This opening spans an area of more than 10.3 Å × 5.5 Å (Fig. 3d), which is sufficiently large for the $\sigma_1$R ligands such as PRE084 (~7.5 Å × 5.5 Å measured transversely) and S1RA (~7.6 Å × 5.2 Å transversely) to pass through. Therefore, the xl$\sigma_1R_{open-endo}$ structure may represent an open-like conformation for $\sigma_1$R and is consistent with PATH2 that predicts the ligand entry between α4 and α5.

Like xl$\sigma_1R_{closed-endo}$, the xl$\sigma_1R_{open-endo}$ crystals were grown in the absence of any known ligand, but the xl$\sigma_1R_{open-endo}$ structure also has an unidentifiable electron density within its ligand-binding pocket (Supplementary Fig. 3b). To investigate if the identified opening between α4 and α5 was induced randomly by the unknown molecule, or whether it is indeed relevant for ligand access to the xl$\sigma_1$R binding site, we further determined two structures for the xl$\sigma_1$R-PRE084 complex in the same crystallization condition as xl$\sigma_1R_{open-endo}$, by either co-crystallizing PRE084 with xl$\sigma_1$R (xl$\sigma_1R_{open-PRE084-co}$, 3.10 Å) or soaking PRE084 directly into the xl$\sigma_1R_{open-endo}$ crystals (xl$\sigma_1R_{open-PRE084-soak}$, 2.85 Å). As expected, both xl$\sigma_1R_{open-PRE084-co}$ and xl$\sigma_1R_{open-PRE084-soak}$ structures are nearly identical to xl$\sigma_1R_{open-endo}$ (Supplementary Fig. 3c), and one PRE084 molecule binds in every protomer of the two structures (Supplementary Fig. 3d). Interestingly, we also observed some dynamics in the Tyr203 side chain in these two PRE084-containing structures, possibly due to their relatively high resolutions. In xl$\sigma_1R_{open-PRE084-co}$, some protomers have Tyr203 in a *gauche+* χ$_1$ rotamer as in xl$\sigma_1R_{open-endo}$, whereas others have Tyr203 in a *trans* χ$_1$ rotamer as in xl$\sigma_1R_{closed-endo}$ (Fig. 3e). In xl$\sigma_1R_{open-PRE084-soak}$, while most protomers have Tyr203 adopt a *gauche+* χ$_1$ rotamer, a few protomers have Tyr203 in both *gauche+* and *trans* χ$_1$ rotamers

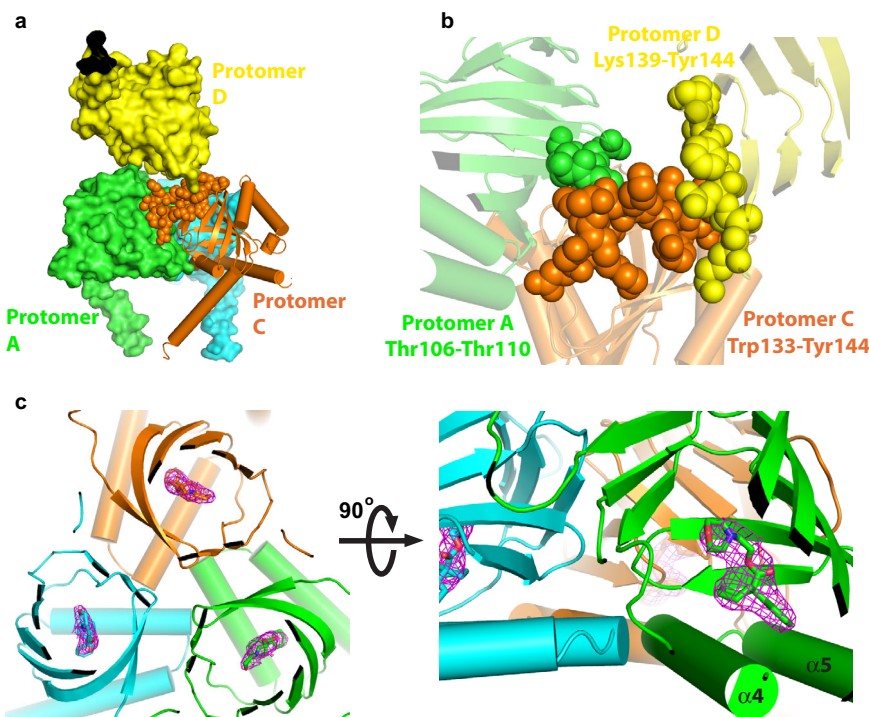

**Fig. 2 Soaking PRE084 into the closed xlσ1R structure. a** Crystal packing of two adjacent xlσ1R$_{closed-endo}$ trimers. Protomers A (in green) and C (in orange) are from the bottom trimer, while Protomer D (in yellow) is from the top trimer, and the other two protomers of the top trimer are not displayed for a clearer view. The tip region of the cupin-fold domain (Trp133–Tyr144) of Protomer C is rendered in spheres. **b** The close-up view of Protomers A (in green), C (in orange), and D (in yellow) from panel (**a**). The residues participating in the close contacts are labeled and are displayed in spheres. **c** Left, one xlσ1R$_{closed-PRE084}$ trimer in the cartoon mode viewed perpendicular to the membrane from the cupin-fold side. Right, the close-up view of the same trimer is viewed parallel to the membrane. The ligand PRE084 is displayed in sticks, and the purple mesh shows the simulated annealing F$_o$–F$_c$ omit map contoured at 3.0$\sigma$ level corresponding to PRE084.

with a partial occupancy for each (Fig. 3f). This result indicates that the σ1R ligand PRE084 can bind σ1R concurrently with an entrance formation between α4 and α5, consistent with the PATH2 hypothesis as a mechanism of ligand entry.

**Blocking the entrance in PATH2 hinders ligand binding in σ1R.** To functionally validate the PATH2 hypothesis of ligand entry to the σ1R binding site, we sought ways to either block the opening between α4 and α5 or prevent its formation, and assess whether either approach would impair ligand binding. The identified opening between α4 and α5 is surrounded by residues mainly from α4/α5, including Pro176, Leu179, Leu183 from α4 and Leu196, Val200, Tyr203, Leu207 from α5, as well as Leu97 from a loop region of the cupin-fold domain (Fig. 4a). These residues outline an 'egg' shape, with Leu183, Leu196, and Val200 at the narrower pointed end, Leu179 and Tyr203 in the middle, and Leu97, Pro176, and Leu207 at the wider rounded end (Fig. 4a). We reasoned that alterations of the middle residues (Leu179 and Tyr203) would probably have the most profound effect on the entrance formation. Therefore, we designed two experiments targeting Leu179 and Tyr 203 to obstruct PATH2-hypothesized ligand entry.

The first idea was to create a disulfide bridge between residues 179 and 203 to prevent the entrance from opening (Supplementary Fig. 4a), leading to a reduction in available binding sites. Therefore, only a fraction of the xlσ1R molecules would bind its ligand (e.g., PRE084), which can be evaluated by stoichiometry measurement using isothermal titration calorimetry (ITC). After introducing a pair of cysteine mutations at residue positions 179 and 203 (xlσ1R-C179/C203), the formation of the disulfide bond

was catalyzed by oxidation. All ITC isotherms measured in this study were best fitted by a one-site model. As expected, the number of the available binding sites per protomer (stoichiometry) in xlσ1R-C179/C203 decreased to 0.18 ± 0.02 after oxidation compared to 0.92 ± 0.01 before oxidation (Fig. 4b and Supplementary Fig. 4b, c), indicating an ~78% efficiency for the disulfide bond formation. Furthermore, the stoichiometry was partially reverted to 0.52 ± 0.02 by re-reducing the oxidized sample of xlσ1R-C179/C203 with β-mercaptoethanol to break the disulfide bond (Fig. 4b and Supplementary Fig. 4d). A partial recovery in the available binding site number after the re-reduction may result from the limited accessibility of β-mercaptoethanol to the disulfide bond between α4 and α5, which are likely covered by detergents in the protein sample due to the hydrophobic exterior face of α4/α5[17]. This data supports that the opening between α4 and α5 mediates ligand entry and binding. Interestingly, the equilibrium dissociation constant ($K_d$) of xlσ1R-C179/C203 for PRE084 also increased by ~9-fold to 6.71 ± 1.44 μM after oxidation compared to 0.68 ± 0.24 μM before oxidation, and was partially reverted to 1.54 ± 0.16 μM after re-reduction (Fig. 4c and Supplementary Fig. 4b–d). These data indicate that the disulfide-bonded xlσ1R-C179/C203 protomers, in which the movement of α4 and α5 is constrained by the disulfide bridge, negatively affected ligand binding in the other, unbonded xlσ1R-C179/C203 protomers. This result is consistent with a previous report that suggests a cooperative ligand binding mechanism in σ1R[20].

To assess the disulfide formation in xlσ1R-C179/C203 more directly, we utilized a bulky reagent, methoxypolyethylene glycol maleimide 5000 (mPEG-Mal-5K), to modify the free thiol groups of Cys179 and Cys203. The only native cysteine (Cys91) was

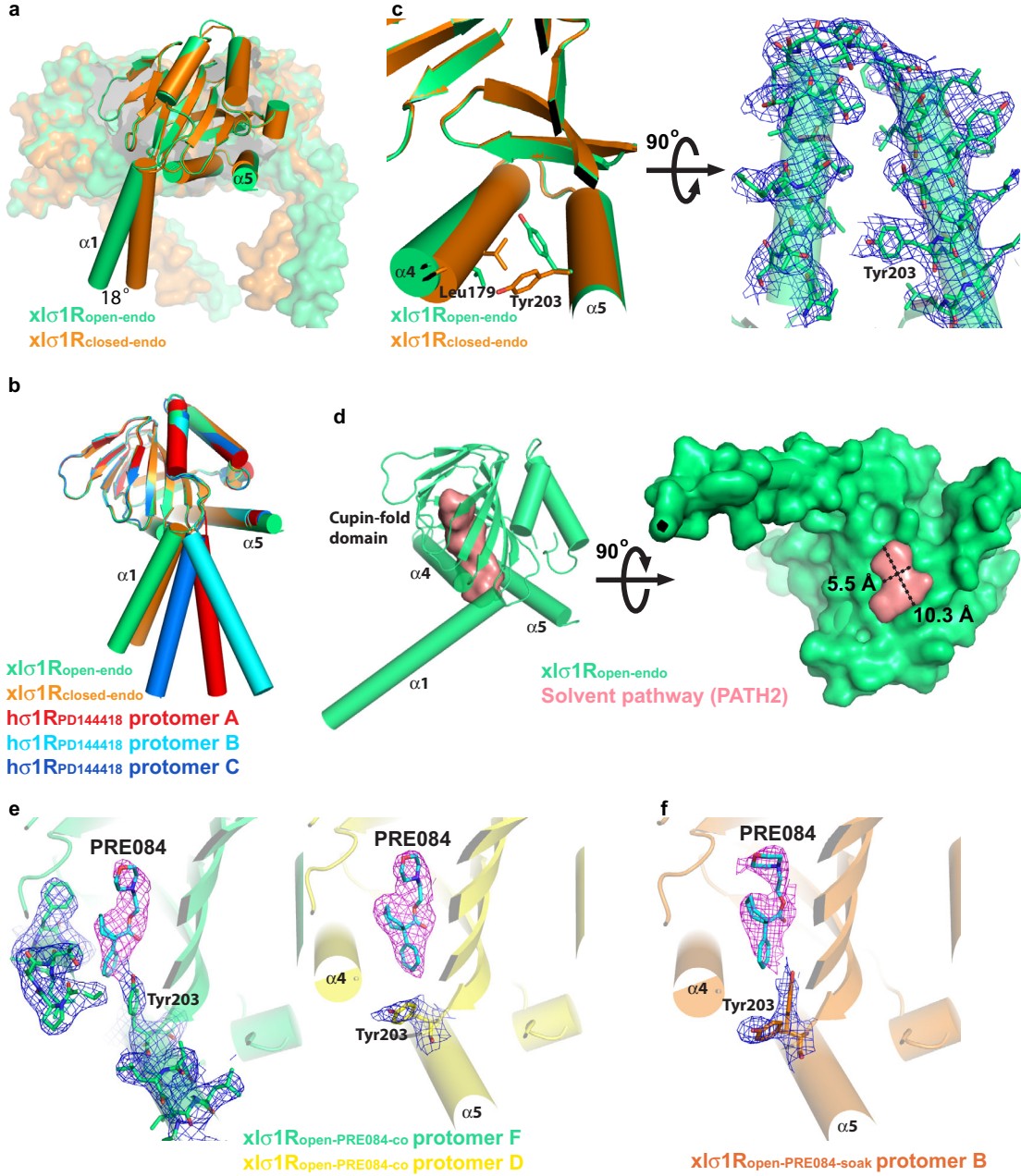

**Fig. 3 An open-like conformation of xlσ1R. a** Superposition of the xlσ1R$_{open-endo}$ trimer (in green) and the xlσ1R$_{closed-endo}$ trimer (in orange), viewed parallel to the membrane. **b** Superposition of one protomer of xlσ1R$_{open-endo}$ (in green), one protomer of xlσ1R$_{closed-endo}$ (in orange), and each protomer (in red, cyan, and blue) of the hσ1R$_{PD144418}$ trimer, viewed parallel to the membrane. **c** Left, the close-up view of the panel (**a**), showing the difference in α4/α5 between the two structures. Leu179 and Tyr203 are rendered in sticks. Right, the electron density map for α4/α5 of the xlσ1R$_{open-endo}$ structure, viewed perpendicular to the membrane from the cupin-fold side. Tyr203 is labeled, and the blue mesh shows the simulated annealing 2F$_o$–F$_c$ map contoured at 1.2σ level. **d** A solvent pathway connecting the ligand-binding site of xlσ1R$_{open-endo}$ and the outside milieu rendered in a pink surface. Left, viewed parallel to the membrane. Right, viewed perpendicular to the membrane from the membrane side. The approximate dimension of the entrance is indicated by a dashed cross. **e**, **f** Protomer F (in green) and D (in yellow) of xlσ1R$_{open-PRE084-co}$, and Protomer B (in orange) of xlσ1R$_{open-PRE084-soak}$ are displayed side-by-side to compare the rotamer of Tyr203. The blue mesh shows the simulated annealing 2F$_o$–F$_c$ map contoured at 1.2σ level, and the purple mesh shows the simulated annealing F$_o$–F$_c$ omit map contoured at 3.0 σ level corresponding to PRE084 (in cyan sticks).

mutated to serine to facilitate this analysis (see "Methods" for details). Modification of mPEG-Mal-5K adds 5 kDa per modification to the modified protein, which causes a shift of the relative mobility of the protein in a sodium dodecyl sulfate-polyacrylamide gel electrophoresis (SDS-PAGE) gel. Considering that the access of mPEG-Mal-5K to Cys179 or Cys203 would likely be impeded by the nearby detergents that cover the hydrophobic exterior face of α4/α5, we expected a relatively

limited efficiency for modification, and therefore this result was only used for correlative analysis. As expected, the modification percentage of xlσ1R-C179/C203 before oxidation reached merely 21.9% (Fig. 4d), though the majority of the receptors are thought to be disulfide-free (Fig. 4b). However, consistent with the disulfide bond formation and breakage, the modification percentage of xlσ1R-C179/C203 decreased to 3.7% after oxidation and returned to 15.9% after re-reduction with β-mercaptoethanol

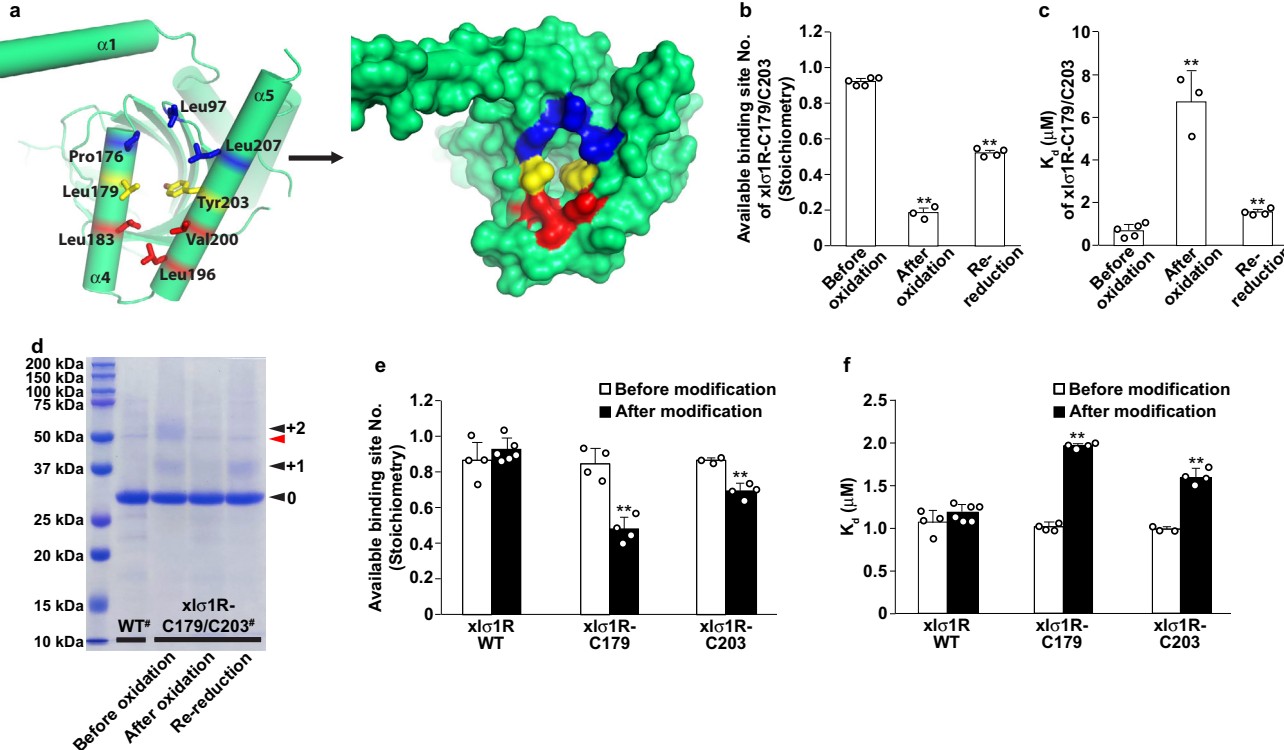

**Fig. 4 Blocking the putative entrance in PATH2. a** One protomer of xlσ1R_open-endo viewed perpendicular to the membrane from the membrane side. Left, the residues surrounding the entrance are displayed in sticks. Right, the surface representation shows an "egg" shape of the entrance residues in three parts: the rounded end (in blue), the middle (in yellow), and the pointed end (in red). **b, c** The stoichiometry (available binding site number per protomer, panel **b**) and the equilibrium dissociation constant ($K_d$, panel **c**) determined by ITC for the xlσ1R-C179/C203 protein in the indicated conditions. All ITC measurements were repeated with biologically independent samples. "Before oxidation": $n = 5$. "After oxidation": $n = 3$. "Re-reduction": $n = 4$. Two-tailed Student's $t$-test was performed between the conditions of "Before oxidation" and "After oxidation", and between the conditions of "After oxidation" and "Re-reduction". The $p$ values are provided in a Source Data file. $**p < 0.01$. **d** Modification of mPEG-Mal-5K to the xlσ1R-C179/C203 protein in the indicated conditions analyzed by SDS-PAGE and Coomassie blue staining. The wild-type (WT) xlσ1R served as the control. The pound sign (#) indicates that the protein contains the extra C91S mutation (see "Methods" for details). The black arrowheads labeled with 0, +1, and +2 indicate the band positions without, with one, and with two mPEG-Mal-5K modification(s). The red arrowhead indicates a minor impurity band near 50 kDa, which does not affect the gel analysis. The modification experiment was repeated three times independently with similar results. **e, f** The stoichiometry (available binding site number per protomer, panel e) and the equilibrium dissociation constant ($K_d$, panel f) determined by ITC for the indicated xlσ1R proteins before and after the mPEG-Mal-5K modification. All ITC measurements were repeated with biologically independent samples. "Before modification": $n = 4$ for xlσ1R WT, $n = 4$ for xlσ1R-C179, $n = 3$ for xlσ1R-C203. "After modification": $n = 6$ for xlσ1R WT, $n = 4$ for xlσ1R-C179, $n = 4$ for xlσ1R-C203. Two-tailed Student's $t$-test was performed between the groups of "Before modification" and "After modification". The $p$ values are provided in a Source Data file. $**p < 0.01$. For panels **b**, **c**, **e**, **f**, data are presented as mean + SD in the bar graphs, which are overlaid with corresponding data points shown as white dots. Source data are provided as a Source Data file.

(Fig. 4d). This result correlates well with the ITC data for xlσ1R-C179/C203.

The second idea was to modify the residue 179 or 203 with a bulky reagent to sterically block the opening between α4 and α5, which would also decrease the number of the available ligand binding sites. To do so, we generated a Leu179-to-cysteine mutant (xlσ1R-C179) and a Tyr203-to-cysteine mutant (xlσ1R-C203), and chose mPEG-Mal-5K as the blocking modifier. This reagent is bulky and has been shown to successfully modify the free thiol group of Cys179 or Cys203 in xlσ1R (Fig. 4d), and has the advantage to allow quick assessment of the relative modification levels for the ITC samples. Since xlσ1R contains a native cysteine (Cys91), we first tested the effect of mPEG-Mal-5K modification of Cys91 on PRE084 binding. The stoichiometry (number of available binding sites per protomer) and affinity ($K_d$) changed little for wild-type xlσ1R after mPEG-Mal-5K treatment (Fig. 4e, f), indicating that the mPEG-Mal-5K modification of Cys91 would not interfere with the following analysis for Cys179 or Cys203. Therefore, Cys91 was not mutated in this analysis. We then measured PRE084 binding by ITC for xlσ1R-C179 or

xlσ1R-C203 after mPEG-Mal-5K modification. As expected, the number of the available binding sites per protomer (stoichiometry) decreased to 0.48 ± 0.07 from 0.85 ± 0.09 for xlσ1R-C179, and to 0.69 ± 0.04 from 0.87 ± 0.01 for xlσ1R-C203 (Fig. 4e). The blockade percentage was ~44% for xlσ1R-C179 and ~21% for xlσ1R-C203, which correlates with the SDS-PAGE analysis with more modification in xlσ1R-C179 (~23%) than in xlσ1R-C203 (~13%) (Supplementary Fig. 4e), possibly due to the location of Cys203 being in a more occluded region compared to Cys179 when the receptor adopts an open conformation (Fig. 4a). Similarly, the equilibrium dissociation constant ($K_d$) increased from 1.02 ± 0.05 μM to 1.95 ± 0.05 μM for the modified xlσ1R-C179, and from 0.99 ± 0.03 μM to 1.61 ± 0.09 μM for the modified xlσ1R-C203 (Fig. 4f), again indicating that the modified xlσ1R protomers impaired ligand binding in the other, unmodified protomers. Meanwhile, we are aware of the limitations associated with the modification experiment, such as the limited accessibility to Cys179 or Cys203, the relatively low efficiency of modification by mPEG-Mal-5K, and the extended shape of the mPEG-Mal-5K molecule that might have affected more xlσ1R protomers than the

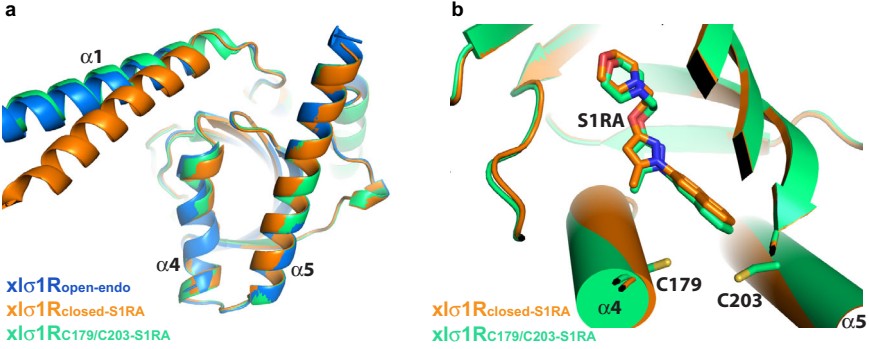

**Fig. 5 The crystal structure of xlσ1R$_{C179/C203-S1RA}$. a** Superposition of one protomer of xlσ1R$_{C179/C203-S1RA}$ (in green), xlσ1R$_{open-endo}$ (in blue), and xlσ1R$_{closed-S1RA}$ (in orange), viewed perpendicular to the membrane from the membrane side. **b** The close-up view of the superposition of one protomer of xlσ1R$_{C179/C203-S1RA}$ (in green) and xlσ1R$_{closed-S1RA}$ (in orange), viewed parallel to the membrane. Cys179, Cys203, and the ligand, S1RA, are displayed in sticks.

one being modified. Therefore, this result needs to be interpreted with caution. Nevertheless, the simplest explanation for the xlσ1R-C179 or xlσ1R-C203 data is that the mPEG-Mal-5K modification physically blocks the entrance between α4 and α5, providing supplementary support for the PATH2 hypothesis of ligand access.

**A structure for the xlσ1R-C179/C203 double mutant**. To further validate the structural integrity of the cysteine mutants of xlσ1R, we solved a structure for the xlσ1R-C179/C203 double mutant (xlσ1R$_{C179/C203-S1RA}$, 3.80 Å) to ensure that the folding of xlσ1R was not disrupted by the cysteine mutation(s). The xlσ1R$_{C179/C203-S1RA}$ structure was solved in the P2$_1$ space group, with 24 protomers in each asymmetric unit (Supplementary Fig. 5a). The same homotrimeric organization was observed in xlσ1R$_{C179/C203-S1RA}$, which resembles closely other xlσ1R structures. Superposition of the xlσ1R$_{C179/C203-S1RA}$ homotrimer and other xlσ1R structures excluding α1 yielded all-atom RMSDs of 0.2–0.3 Å (Fig. 5a). The α4/α5 region of xlσ1R$_{C179/C203-S1RA}$ appeared also similar to the closed and the open-like conformations (Fig. 5a). The xlσ1R$_{C179/C203-S1RA}$ crystals were grown in the presence of PRE084, and S1RA was later soaked in after the crystals had formed. Each protomer of the xlσ1R$_{C179/C203-S1RA}$ structure contained one S1RA molecule in the ligand-binding pocket, which adopts a similar pose to the ligand in the xlσ1R$_{closed-S1RA}$ structure (Fig. 5b). As expected, no disulfide bond was formed between Cys179 and Cys203 in this structure (Fig. 5b), thus allowing S1RA access to and binding in the ligand-binding site. This result indicates that substitution of the residues 179 and 203 with cysteine did not perturb the structural integrity of xlσ1R, and the resulting xlσ1R mutants retained the ligand-binding activity.

## Discussion

The combinatory approach of structural and functional studies has gained critical insight into the molecular underpinnings of σ1R function. However, how ligands access the ligand-binding site in σ1R remains elusive. In this study, we focused on this important question and interrogated two ligand entry pathways proposed for σ1R, PATH1 and PATH2, and our structural and functional data favor the PATH2 hypothesis for ligand access. Moreover, our data offered some new information for further discussion.

First, within the ligand-binding pocket of xlσ1R$_{closed-PRE084}$ or xlσ1R$_{closed-S1RA}$, the ligand is coordinated through a salt bridge between its cationic nitrogen and the carboxy group of Glu169 (the equivalent of Glu172 in hσ1R), and through hydrophobic interactions with Val81, Trp86, Met90, Tyr100, Leu102, Phe104,

Tyr117, Ile121, Trp161, Ile175, Leu179, Phe181, Ala182, and Tyr203 (Supplementary Fig. 6), which is nearly identical to the ligand-binding pattern in hσ1R[17]. The highly conserved structures of σ1R and their ligand binding sites from two different species further support the single-transmembrane-helix model[16] for σ1R as revealed by the crystal structures[17,20].

Second, the structures of the xlσ1R-PRE084 complex in both closed and open-like conformations provide a unique opportunity to discuss how a ligand may enter and exit σ1R, which are likely reversed processes. Superposition of xlσ1R$_{closed-PRE084}$ and xlσ1R$_{open-PRE084-co}$ indicates that PRE084 adopts slightly different poses in these two states (Supplementary Fig. 7), and it is feasible to propose the following exit process. From the closed state to the open-like state, the phenyl ring of PRE084 moves ~2 Å closer to α4 to avoid a steric clash with the side chain of Tyr203 (α5), which points toward the ligand-binding site. As a result, the PRE084 phenyl ring pushes against part of α4, causing it to rotate and move slightly away from α5. Interestingly, the steric pressure between PRE084 and α4 has also been observed in previous simulations[20,22]. Afterward, PRE084 may exit σ1R through the opening formed between α4 and α5. Similarly, PRE084 may enter σ1R in an inversed process. However, though we have observed both closed and an open-like conformation for σ1R, the mechanism that regulates the opening and closing of σ1R remains unknown. Meanwhile, the conformational change of α4 has also been suggested in σ1R agonism[20], but its relationship with the conformational changes that lead to the opening of the receptor remains to be investigated.

Third, the PATH2 access hypothesis infers that the σ1R ligand enters from, and exits to, the membrane rather than an aqueous environment (Figs. 1a, 1b). Given the largely hydrophobic nature of most synthetic σ1R ligands[2], it would be reasonable to assume that the ligand would partition into the membrane, or at the surface of the membrane (as the ligand becomes amphipathic when protonated). Since σ1R is primarily localized within cells, it has been suggested that a synthetic ligand would need to partition into and pass through, at least the plasma membrane[22]. Drug delivery studies also show that hydrophobic molecules do partition and enrich into the cellular membranes[23]. Therefore, a membrane pathway would not be a prohibitory factor for ligand entry and exit in σ1R.

Fourth, one major difference among the σ1R structures is the orientation of the transmembrane helix α1 (Fig. 3b). It is interesting to speculate whether the different orientation bears physiological relevance. Several residues from α1, including Trp26 and Leu27, form hydrophobic interactions with a loop region of the cupin-fold domain (Supplementary Fig. 8a), including Leu97,

which is like a plug between α4 and α5 and is one of the residues surrounding the putative ligand entrance (Fig. 4a). Since α1 pulls away from Leu97 in the open-like conformation of xlσ$_1$R (Fig. 3a), it is tempting to postulate that the Leu97-containing loop may have space to swing slightly away from the membrane (Supplementary Fig. 8b), so that a ligand dissolved at the membrane surface may slide in and out between α4 and α5 to access the ligand-binding site from within the membrane (Supplementary Fig. 8b). Unfortunately, this structural feature may be a very dynamic process and was not captured in the open-like xlσ$_1$R structures in this study.

Finally, it is worth mentioning another structural effort that we attempted to capture an open conformation of σ$_1$R by soaking a ligand of extended length into the xlσ$_1$R crystals. We anticipated that the ligand would extend through the ligand entrance and cause the receptor to adopt an open state. Such an extended ligand, N$^1$,N$^3$-bis(1-((R)-3-([1,1′-biphenyl]-4-yl)butyl)-piperidin-4-yl)malonamide (DIM-3C)[21], was kindly provided by Dr. Collina and Dr. Rossino for soaking the xlσ$_1$R crystals. Unfortunately, the highly hydrophobic DIM-3C could only be dissolved with organic solvents such as methanol and dimethyl sulfoxide, which damaged the xlσ$_1$R crystals, and no structure was obtained for the DIM-3C soaking experiment. However, interestingly, in one of the 24 protomers of the xlσ$_1$R$_{C179/C203-S1RA}$ structure, an extra electron density resembling the shape of S1RA was observed near the ligand entrance in addition to the S1RA molecule in the ligand-binding site (Supplementary Fig. 5b). The density was readily and best modeled with an S1RA molecule. Its morpholine ring sits between α4 and α5 and is surrounded by residues 179, 203, and Leu97, whereas the rest of the ligand remains outside of the receptor, possibly representing an entering (or exiting) pose for the ligand. Meanwhile, due to the relatively low resolution (3.80 Å) of the xlσ$_1$R$_{C179-S1RA}$ structure, this extra density needs to be interpreted with caution, and we used it only for supplementary discussion.

## Methods

**Chemicals**. All chemicals in this study were purchased from Sigma-Aldrich (St. Louis, MO) except 4-[2-((5-methyl-1-(naphthalene-2-yl)-1H-pyrazol-3-yl)oxy) ethyl]morpholine (S1RA) hydrochloride, which was purchased from Topscience Co. Ltd. (Shanghai, China).

**Protein expression and purification**. The gene encoding the full-length wild-type σ$_1$R from *Xenopus laevis* (xlσ$_1$R, NCBI accession NP_001087013.1) was synthesized by Genewiz (Suzhou, China) in a modified pPICZ plasmid (Thermo Fisher Scientific, Waltham, MA) containing an amino-terminal tag of decahistidine and Tobacco etch virus (TEV) protease recognition site following the hemagglutinin signal peptide. Cysteine mutations were introduced by site-directed mutagenesis using QuikChange II system (Agilent, Santa Clara, CA) according to the manufacturer's recommendation, and all mutations were verified by sequencing. The wild-type xlσ$_1$R and mutants were overexpressed in yeast strain GS115 (*Pichia pastoris*) cells by adding 1% (v/v) methanol and 2.5% (v/v) dimethyl sulfoxide (DMSO) at OD$_{600\ nm}$ of ~1 and shaking at 20 °C for 48 h. Cell pellets were resuspended in lysis solution (LS) containing 20 mM Tris-HCl pH 7.5, 150 mM NaCl, 10% (v/v) glycerol, 1 mM phenylmethanesulfonyl fluoride, and 2 mM β-mercaptoethanol, and lysed by an ATS AH-1500 high-pressure homogenizer (Shanghai, China) at 1300 MPa. Protein was extracted by addition of 1% (w/v) n-dodecyl-β-D-maltopyranoside (DDM, Anatrace, Maumee, OH) and 0.1% (w/v) cholesteryl hemisuccinate (CHS, Anatrace, Maumee, OH) at 4 °C for 2 h and the extraction mixture was centrifuged at 200,000 × g for 20 min at 4 °C. The supernatant was then loaded onto a cobalt metal affinity column, washed with 20 bed-volume of LS containing 3 mM DDM, 0.01% (w/v) CHS, and 20 mM imidazole pH 8.0, and eluted with LS supplemented with 3 mM DDM, 0.01% (w/v) CHS and 250 mM imidazole pH 8.0.

**Crystallization**. Affinity-purified xlσ$_1$R was concentrated to 6-8 mg/ml and loaded onto a Superdex 200 Increase 10/300 GL column (GE Healthcare, Chicago, IL) equilibrated in 20 mM Tris-HCl pH7.5, 150 mM NaCl, 5 mM β-mercaptoethanol, 40 mM octyl-β-D-glucopyranoside (OG) and 0.001% (w/v) CHS and was further purified by size-exclusion chromatography (SEC). SEC-purified xlσ$_1$R was then

concentrated to 5–6 mg/ml as approximated by ultraviolet absorbance, and 500 nl of protein solution was mixed with an equal volume of crystallization solution manually in a vapor diffusion sitting-drop setup and was incubated at 20 °C. (1) The xlσ$_1$R$_{closed-endo}$ crystals grew in 0.1 M NaCl, 0.1 M sodium citrate pH 5.1, 26% (v/v) PEG 400, and 10 mM KAu(CN)$_2$. (2) The xlσ$_1$R$_{closed-PRE084}$ crystals grew in 0.1 M NaCl, 0.1 M sodium citrate pH 5.2, 28% (v/v) PEG 400, 10 mM KAu(CN)$_2$, and were soaked with 0.4 mM [2-(morpholin-4-yl)ethyl] 1-phenylcyclohexane-1-carboxylate (PRE084) hydrochloride. (3) The xlσ$_1$R$_{closed-S1RA}$ crystals grew in 0.1 M NaCl, 0.1 M sodium citrate pH 5.2, 26% (v/v) PEG 400, 10 mM KAu(CN)$_2$, and were soaked with 0.4 mM S1RA hydrochloride. (4) The xlσ$_1$R$_{open-endo}$ crystals grew in 0.1 M sodium HEPES pH 6.5, and 28% PEG 300. (5) The xlσ$_1$R$_{open-PRE084-co}$ crystals grew in 0.1 M sodium HEPES pH 6.5, 28% PEG 300, and 0.4 mM PRE084 hydrochloride. (6) The xlσ$_1$R$_{open-PRE084-soak}$ crystals grew in 0.1 M sodium HEPES pH 6.5, 28% PEG 300, and were soaked with 0.4 mM PRE084 hydrochloride. (7) The xlσ$_1$R$_{C179/C203-S1RA}$ crystals grew in 0.05 M ADA pH 6.5, 24% PEG 400, 1 mM octyl-maltoside fluorinated, 0.4 mM PRE084 hydrochloride, and were soaked with 2 mM S1RA hydrochloride. The xlσ$_1$R crystals usually appear within a week, and reach full-size in two weeks. The crystals were cryo-protected by raising the precipitant concentrations (final 36% for PEG 400 and 32% for PEG 300) with a 2% (v/v) incremental step, and flash-frozen in liquid nitrogen.

**Data collection, structure solution, and structural analysis**. Diffraction data were collected on beamlines BL18U1 and BL19U1[24] of the National Facility for Protein Science in Shanghai (NFPS) at the Shanghai Synchrotron Radiation Facility (SSRF). The data were indexed, integrated, and scaled using the autoPROC pipeline package (Global Phasing Limited)[25], which includes XDS[26] and AIMLESS (CCP4 package)[27]. All xlσ$_1$R structures were solved by molecular replacement with Phaser[28] using the published hσ$_1$R$_{PD144418}$ structure (PDB entry 5HK1) as a template. Manual model building and refinement were carried out using Coot[29] and phenix.refine[30], and Molprobity[31] was used to monitor and improve protein geometry. The asymmetric unit of all xlσ$_1$R structures contains 12 or 24 protomers, and so the non-crystallographic symmetry (NCS) was applied throughout the refinement to improve the map, except that for xlσ$_1$R$_{open-PRE084-co}$ (3.10 Å) and xlσ$_1$R$_{open-PRE084-soak}$ (2.85 Å), the NCS restraints were relaxed in the last a few rounds of refinement. The data collection and refinement statistics were generated using phenix.table_one[30] and the values are listed in Table 1. All structural figures, RMSD calculations, and length measurements were performed in PyMOL (Schrödinger, LLC). Accessibility analysis was performed using the volume-filling program HOLLOW[32] with default settings.

**Oxidized disulfide formation and re-reduction**. The disulfide bond formation between Cys179 and Cys203 was catalyzed by adding a freshly prepared CuSO$_4$ stock in the xlσ$_1$R-C179/C203 protein sample, which was SEC-purified in the absence of any reducing reagent, to a final concentration of 200 μM for 1 h at the room temperature. The oxidation reaction was stopped by the addition of 10 mM EDTA pH 8.0 for 10 min at room temperature, and the protein sample was re-purified by SEC. For the re-reduction experiment, the oxidized xlσ$_1$R-C179/C203 sample was incubated with 60 mM β-mercaptoethanol at 4 °C for 12 h, and the protein sample was then re-purified by SEC. The oxidized or re-reduced xlσ$_1$R-C179/C203 samples were subjected to ITC or SDS-PAGE analysis.

**Cysteine modification**. Modification of wild-type and cysteine mutants of xlσ$_1$R was performed with a bulky thiol-modifying reagent, methoxypolyethylene glycol maleimide 5000 (mPEG-Mal-5K)[33]. Briefly, the SEC-purified xlσ$_1$R protein in the absence of β-mercaptoethanol were incubated with 1 mM mPEG-Mal-5K at room temperature for 1 h, and the modification reaction was stopped by addition of β-mercaptoethanol to a final concentration of 10 mM before another SEC purification to remove any free mPEG-Mal-5K. For the entrance blocking experiment, the modified xlσ$_1$R-C179 or xlσ$_1$R-C203 proteins were subjected to ITC or SDS-PAGE analysis. For the assessment of disulfide formation in xlσ$_1$R-C179/C203, the modified sample was analyzed by SDS-PAGE. To reduce background modification and facilitate gel analysis, the only native cysteine (Cys91) was mutated to serine in all samples for the disulfide analysis.

**Band intensity analysis of SDS-PAGE gels**. Protein bands in SDS-PAGE gels were visualized by Coomassie blue staining, and the band intensities were quantified using the ImageJ software[34]. Briefly, the gel image was first converted to grayscale, and equal-sized rectangles were drawn to encompass the bands of interest to measure their intensities. The intensity of a background rectangle was also measured for background correction. The modification ratio of the xlσ$_1$R variants was calculated by the following equation: modification ratio (%) = (modified band intensity)/(modified + unmodified band intensities) × 100.

**Isothermal titration calorimetry (ITC)**. The ITC measurement for the binding of PRE084 to xlσ$_1$R was performed using a MicroCal iTC200 microcalorimeter (Malvern Instruments, Malvern, UK). The xlσ$_1$R samples and PRE084 were prepared in the titration buffer containing 20 mM sodium HEPES pH 7.5,

**Table 1 Data collection and refinement statistics for the $xl\sigma_1R$ structures.**

| | $xl\sigma_1R_{closed-endo}$ | $xl\sigma_1R_{closed-PRE084}$ | $xl\sigma_1R_{closed-SIRA}$ | $xl\sigma_1R_{open-endo}$ | $xl\sigma_1R_{open-PRE084-co}$ | $xl\sigma_1R_{open-PRE084-soak}$ | $xl\sigma_1R_{C179/C203-SIRA}$ |
|---|---|---|---|---|---|---|---|
| **PDB ID** | 7W2B | 7W2C | 7W2D | 7W2E | 7W2F | 7W2G | 7W2H |
| *Data collection* | | | | | | | |
| Space group | P 1 2$_1$ 1 | P 1 2$_1$ 1 | P 1 2$_1$ 1 | P 2$_1$2$_1$2$_1$ | P 2$_1$2$_1$2$_1$ | P 2$_1$2$_1$2$_1$ | P 1 2$_1$ 1 |
| Wavelength (Å) | 0.9793 | 0.9792 | 0.9792 | 0.9793 | 0.9792 | 0.9785 | 0.9785 |
| Unit cell | | | | | | | |
| $a, b, c$ (Å) | 86.7, 148.5, 172.1 | 87.2, 148.8, 172.8 | 86.3, 147.9, 171.4 | 135.1, 161.5, 200.2 | 135.3, 161.0, 202.2 | 135.0, 161.0, 201.9 | 134.5, 200.8, 160.6 |
| $\alpha, \beta, \gamma$ (°) | 90, 92.4, 90 | 90, 92.8, 90 | 90, 91.7, 90 | 90, 90, 90 | 90, 90, 90 | 90, 90, 90 | 90, 90.1, 90 |
| Resolution (Å) | 3.20 (3.26–3.20) | 3.33 (3.39–3.33) | 3.47 (3.53–3.47) | 3.56 (3.62–3.56) | 3.10 (3.15–3.10) | 2.85 (2.90–2.85) | 3.80 (3.86–3.80) |
| Unique reflections | 70,446 (3533) | 59,588 (3209) | 53,754 (2755) | 51,558 (2562) | 80,396 (3993) | 102,980 (5102) | 67,737 (3410) |
| Multiplicity | 6.9 (7.2) | 6.6 (6.3) | 3.4 (3.3) | 4.6 (4.6) | 5.2 (5.4) | 11.2 (11.6) | 2.6 (2.4) |
| Completeness (%) | 98.3 (100.0) | 92.8 (100.0) | 96.5 (99.9) | 97.4 (98.2) | 99.5 (99.8) | 99.9 (100.0) | 81.1 (81.9) |
| $I/\sigma I$ | 12.2 (2.0) | 11.2 (2.4) | 6.7 (1.7) | 7.7 (1.8) | 8.1 (1.6) | 13.3 (1.8) | 4.8 (1.9) |
| $R_{merge}$ | 0.137 (1.129) | 0.146 (0.859) | 0.147 (0.786) | 0.141 (0.754) | 0.135 (0.931) | 0.134 (1.505) | 0.159 (0.431) |
| $R_{meas}$ | 0.148 (1.216) | 0.159 (0.932) | 0.175 (0.942) | 0.158 (0.843) | 0.151 (1.033) | 0.140 (1.574) | 0.188 (0.465) |
| $R_{pim}$ | 0.056 (0.450) | 0.061 (0.360) | 0.094 (0.513) | 0.069 (0.367) | 0.065 (0.441) | 0.041 (0.457) | 0.110 (0.313) |
| $CC_{1/2}$ | 0.996 (0.735) | 0.993 (0.780) | 0.988 (0.652) | 0.998 (0.822) | 0.998 (0.774) | 0.999 (0.808) | 0.989 (0.853) |
| *Refinement* | | | | | | | |
| Resolution (Å) | 3.20 (3.25–3.20) | 3.33 (3.39–3.33) | 3.47 (3.53–3.47) | 3.56 (3.62–3.56) | 3.10 (3.14–3.10) | 2.85 (2.88–2.85) | 3.80 (3.85–3.80) |
| No. reflections | 70,305 | 59,445 | 53,659 | 51,418 | 80,119 | 102,713 | 67,528 |
| Completeness (%) | 98.1 | 92.6 | 96.3 | 97.1 | 99.2 | 99.6 | 80.8 |
| $R_{work}/R_{free}$ (%) | 24.1/26.9 | 22.9/25.9 | 23.9/26.7 | 26.4/29.7 | 25.6/28.8 | 25.4/28.3 | 28.3/31.8 |
| No. of atoms | 20,673 | 20,833 | 20,593 | 20,974 | 20,904 | 20,822 | 40,341 |
| Protein | 20,469 | 20,472 | 20,269 | 20,959 | 20,333 | 20,224 | 39,716 |
| Ligands | 204 | 360 | 324 | 15 | 416 | 416 | 625 |
| Solvent | | 1 | | | 155 | 182 | |
| Average B-factor | 85.73 | 91.87 | 104.5 | 101.3 | 80.54 | 76.89 | 82.30 |
| Protein | 85.83 | 91.91 | 104.7 | 101.3 | 80.40 | 76.90 | 82.36 |
| Ligands | 75.06 | 89.49 | 92.28 | 76.80 | 95.84 | 82.54 | 78.22 |
| Solvent | | 58.30 | | | 58.11 | 63.23 | |
| *Ramachandran* | | | | | | | |
| Favored (%) | 98.01 | 97.39 | 99.01 | 98.60 | 98.59 | 98.98 | 97.72 |
| Allowed (%) | 1.99 | 2.61 | 0.91 | 1.40 | 1.41 | 1.02 | 2.24 |
| Outliers (%) | 0.00 | 0.00 | 0.08 | 0.00 | 0.00 | 0.00 | 0.04 |
| RMS bonds (Å) | 0.002 | 0.002 | 0.002 | 0.003 | 0.002 | 0.003 | 0.002 |
| RMS angles (°) | 0.537 | 0.495 | 0.575 | 0.559 | 0.449 | 0.694 | 0.456 |
| Clashscore | 3.50 | 3.04 | 4.97 | 4.82 | 4.16 | 3.72 | 4.28 |

Statistics for the highest-resolution shell are shown in parentheses.

150 mM NaCl, 1 mM DDM, and 0.005% (w/v) CHS. The first injection was 0.2 µl in volume and the subsequent injections were 2.0 µl. Before the data collection, the system was equilibrated to 25 °C with the stirring speed set to 750 rounds per min. The titration curve of PRE084 was generated with nineteen 2.0-µl injections at 90-s intervals. All the $xl\sigma_1R$ titration data were best fitted by a one-site binding model using ORIGIN 7, and calculations were performed using Microsoft Excel for Mac 2016. The values of the available binding site number per protomer (stoichiometry) and the equilibrium dissociation constant ($K_d$) were determined from the average of three to six biologically independent ITC measurements and were expressed as mean ± SD in the text. Two-tailed Student's t-test was performed for statistical analysis, and details were described in relevant figure legends.

**Reporting summary**. Further information on research design is available in the Nature Research Reporting Summary linked to this article.

## Data availability

The atomic coordinates and structure factors of the $xl\sigma_1R$ structures generated in this study have been deposited in the Protein Data Bank under the following accession codes: 7W2B ($xl\sigma_1R_{closed-endo}$), 7W2C ($xl\sigma_1R_{closed-PRE084}$), 7W2D ($xl\sigma_1R_{closed-S1RA}$), 7W2E ($xl\sigma_1R_{open-endo}$), 7W2F ($xl\sigma_1R_{open-PRE084-co}$), 7W2G ($xl\sigma_1R_{open-PRE084-soak}$), 7W2H ($xl\sigma_1R_{C179/C203-S1RA}$). A previously reported $h\sigma_1R$ structure complexed with the antagonist PD144418 ($h\sigma_1R_{PD144418}$) used in this study is available in the Protein Data Bank under accession code 5HK1. Source data are provided with this paper.

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

## Acknowledgements

Diffraction data used in this study were collected on beamlines BL18U1 and BL19U1 of the National Facility for Protein Science in Shanghai (NFPS) at Shanghai Synchrotron Radiation Facility (SSRF). The authors thank the staff from these beamlines for assistance during data collection. We thank the Center of Growth, Metabolism, and Aging at Sichuan University for granting access to a MicroCal iTC200 microcalorimeter for ITC measurements. The authors also thank Dr. Simona Collina and Dr. Giacomo Rossino at the University of Pavia for kindly providing DIM-3C for soaking the $xl\sigma_1R$ crystals and thank Dr. Matthias Quick at Columbia University for critical reading and suggestions of the manuscript. This work was supported in part by the National Natural Science Foundation of China (NSFC) grant 31770783, and the 1.3.5 Project for Disciplines of Excellence grant by West China Hospital of Sichuan University to XZ.

## Author contributions

X.Z. and Z.S. conceived the project and wrote the paper. F.M. performed the structural studies; F.M., Y.X., and Y.J. performed the functional studies; and all authors analyzed the data.

## Competing interests

The authors declare no competing interests.
