## [Peer Review File · Nature Communications]

An open-like conformation of the sigma-1 receptor reveals its ligand entry pathwayREVIEWER COMMENTS

Reviewer #1 (Remarks to the Author):

The manuscript by Meng et al titled “An open conformation of sigma-1 receptor” reports crystal structures of xenopus sigma-1 receptor in complex with either a known agonist PRE-84 or an endogenous unidentified ligand. The first structure is similar to the previously reported structure of human S1R save for small changes in the TM helix, but the second structure shows a different conformation of the two C-terminal helices so that the access to the ligand binding pocket becomes obvious. The authors then modified the residues on the C-terminal helices and showed that agonist binding is compromised. Because the C-terminal helices were thought to form a tight bundle through the “hydrophobic zipper”, previous studies have assumed that the two would stay together during the agonist binding process. The new structure provides a fresh perspective on what structural changes are possible and thus is highly significant. The study provides a simple interpretation of ligand access to its binding site on S1R, and because S1R is involved in multiple diseases and is a drug target, these results will have a significant impact.

The diffraction data and the refinement of structural models are of good quality with statistics expected for the modest resolution of the data. It would be necessary to show the regions of the electron density map corresponding to the beta roll and the helices in supplementary figures to provide confidence on structures.

I appreciate authors effort to validate the conclusions from the structure with functional analyses. However, I am concerned that modification by the mPEG-Mal-5k has low efficiency and that there are higher molecular weight bands on the gel. For example, the lane marked as L182C A has a spear of high molecular weight bands. I suggest that the authors consider these points, although #3 and #4 are not meant as a demand for additional experiments.

1. Is it possible that the endogenous ligand prevents the modification? If so, perhaps extensive dialysis would empty the binding pocket and enhance the modification.
2. The authors should clearly state what is the expected behavior for an xIS1R modified with mPEG-Mal-5k at positions 179, 182, or 203. Does the loss of binding site N correlate with the percentile of modification? And finally, why is there a two fold change in Kd after the modification if only a small fraction of S1R is modified?
3. Could the authors find or synthesize a derivative of a known agonist that would extend between a4 and a5? This would strengthen the conclusion from the structure.
4. Given that single cysteine mutations are functional, would it make sense to make a double cysteine mutation, for example, C179/C203, and test whether ligand binding is affected in the presence and

absence of cys bridge? Or alternatively, whether the formation of a cys bridge is affected by the presence of a ligand?

Overall, the manuscript is clear and has a good logical thread. However, there are multiple grammatical errors that I do not expect to find in a manuscript with an inspiration for Nature Communications. Professional editing or enlisting help from a native English speaker may solve the problem. Finally, I would like to suggest that the authors consider changing S1R_{no-adding} to S1R_{endo} for S1R with an endogenous ligand.

Reviewer #2 (Remarks to the Author):

Meng et al. reports crystal structures of sigma1 receptor with PRE084 and unknown ligand carried over from purification step. Given previously reported crystal structure of human S1R displays a closed conformation in which ligand is occluded in the cupin-fold, it has been questioned how ligand enters to its binding site. The authors claim that previously proposed open mechanism from MD simulation is unlikely to occur, and analyzed 3.56Å resolution crystal structure in an open state but bound unknown substrate in it. In this structure, Y203 side chain points to the intrinsic ligand, and as a consequence, binding pocket becomes solvent-accessible. It is therefore unclear whether the proposed “gate (Y203-L179)” is a canonical ligand entry site or just induced by the binding of unknown ligand.

However, they fall short seriously in their presentation, and the functional evidences are not conclusive to show L179-Y203 gating.

Major points

It is unclear how three monomers in the trimer are similar or identical to each other. According to the crystal symmetry, each three monomers are not necessarily identical. If these are identical, authors could apply NCS to improve the density map. The process was not described in either Methods or main text. If monomers show different conformation, structures of three monomers must be compared, especially around gate structure.

In the relatively low-resolution map, side chain rotamers are not unequivocally determined. Author should present electron density maps for not only particular residues, but also whole molecule especially around gate position to show how Y203 rotamer is defined in the structure.

Figures are not well organized and drawn. Color codes are confusing. For example, in Fig1d, alpha4 helix and some beta-sheets are drawn in orange, but this is not indicated in the figure caption. In Fig. 1a left, author used green and magenta for the position to disrupt in the previous simulation, and similar green and pink colors are used for the to the right panel.

In the present crystal structure, the alpha1 helix is significantly tilted along with the membrane plane, which authors mentioned artifact due to crystal packing. Compared with previous crystal structure in which alpha1 helix locate enough far from alpha4-5 with 2 or 3 OLC molecule in between, present alpha1 seems closer to alpha4-5. This is not clear from the figure. Does alpha1 helix asserts any effect on the binding pocket especially via alpha4-5 helices?

It is unclear that the binding site in the S1Rno-adding is fully open to the solvent. It seems that the gate (L179-Y203) is facing to the membrane surface. Furthermore, even if Y203 took different rotamer conformation, the loop structure between beta-sheet (S99,L100,S101 in human receptor 5hk1, see figure) blocks ligand dissociation. In the figures author provide (especially Fig3d) are only showing part of pocket, and thus unfair and misleading.

*Attached Figure human S1R (pdb:5hk1)

L182 and Y206: magenta, S99, L100 and S101: Yellow (pdb 5hk1)

It is unclear why the binding capacity of human receptor is around 0.5 in contrast to xenopus receptor (~1.0).

Label ratio of used sample must be determined and compared to the binding experiment. Judging from SDS-PAGE, label ratio of human receptor seems less than 20% but the effect of ITC analysis seems too large.

Why affinity changes upon labeling? If L179 or Y203 act as a gate, modification with such a huge reagent may lost its binding ability, which reflect reduced number of binding site (Fig4f). In other words, unlabeled receptor must be healthy and accordingly affinity of receptor that accept ligand must be the same before and after modification. Is it an allosteric effect?

From the kinetic analysis in ref 9, ligand-independent conformational change is proposed. If L179-Y203 is a gate, their simple mutation (alanine or glycine) may affect ligand binding kinetics.

Related to this comment, does Cys replacement of highly conserved L179 and Y203 exert any difference in ligand binding or its kinetics?

L182 and Y206: magenta, S99, L100 and S101: Yellow (pdb 5hk1)

Reviewer #3 (Remarks to the Author):

The manuscript presented by Meng et al. describes two crystal structures of S1R from *Xenopus laevis* (xS1R). The first structure was obtained in the presence of the agonist PRE-084, exhibiting a conformational state consistent with a previously reported human S1R structure bound to (+)-pentazocine. The main difference is observed into the transmembrane helix alpha1, which is attributed to its flexibility or crystal packing. The second structure was crystallized in the absence of any known S1R ligand; however, the authors found an “unknown molecule” presumably from an endogenous origin located at the ligand-binding pocket. A structural comparison between the two xS1R crystallized structures revealed a conformational change in two alpha-helices at the C-terminus (alpha4-alpha5), which is attributed to an apparent opening of the ligand-binding pocket towards the ER membrane. Two residues were identified as key in the opening process (i.e. L179 and Y203), which were mutated by cysteine affecting the ligand-binding properties in S1R.

This work could represent a step forward understanding the conformational changes associated with the ligand binding in the Sigma 1 receptor expressed in different organisms; however, not knowing the chemical nature of the molecule responsible of the conformational changes in xS1Rno-adding makes this work unconvincing. The major concern is about the conclusion that an “open structure” of S1R has been found. The resolution of the crystal structure and the nonspecificity of the bound ligand makes not clear that the structure reported here is in open state. Further studies are required to identify the molecule bound to the ligand-binding pocket in this conformation and to demonstrate that this binding event is not a crystallization artifact.

Authors stated that the current resolution of xS1Rno-adding (which is very low 3.56 Å) limited their ability to identify the bound molecule. However, they consider that this resolution was enough to identify the residues Tyr203 and Leu179 shifted generating an open conformation. At first glance, the xS1Rno-adding structure does not show a significant opening of the alpha4-alpha5 helices to accommodate ligands entering. A structural analysis exhibiting the dimensions of this cavity in the occluded and open structure would be necessary to rationalize the passing of known S1R ligands. Due to the remarkable plasticity in ligand recognition of S1R, ligands from different size and shape should penetrate the gate formed by the Leu179-Tyr203 pair.

To evaluate the importance of the residues Leu179 and Tyr203, cysteine mutations were performed, and ligand-binding properties were evaluated by isothermal titration calorimetry (ITC). Although some effects were observed, it is questionable that these assays are enough to demonstrate that modifying these positions “blocks or impedes ligand entering through the opening”. L179 and Tyr203 have shown to interact with ligands; therefore, it could be trivial to find an effect when they are replaced by Cys modified with a bulky reagent. In this regard, ITC does not demonstrate that “a blockade near residues 179/203 hinders ligand access and binding to xS1R” by means of a mechanism where Leu179-Tyr203 act as a gate. In addition, authors said: “Although wild-type xS1R contains a native cysteine at residue 91

(Cys91), modification of mPEG-Mal-5K on Cys91 showed little influence on binding of the agonist PRE-084 to the wild-type xS1R sample and yielded comparable active binding site numbers". It could be because the side chain of C91 is not oriented to the ligand-binding site.

Minor points:

Authors point out that "a computational simulation further suggests that the opening of S1R occurs in a two-step conformational change that requires disruption of the cupin-fold." It should be noted that recently another computational study (J. Chem. Inf. Model. 2020, 60, 756–765) explored two ligand pathways to enter and exit the binding site. According to this study, the most favored pathway implies deformation of the cupin-fold and the less favored pathway involves a gap between the two membrane-adjacent helices D and E (alpha4 and alpha5). This last pathway has common characteristics with the "open structure" reported in this manuscript. Authors should discuss about this antecedent.

On the other hand, Yano et al (Neuropharmacology 2018, 133, 264-275) reported molecular modeling of several ligands interacting with S1R, including PRE-084. In this work, PRE-084 showed an unique kinetics for entering the ligand binding site. Authors should discuss whether structural characteristics are comparable with the xS1R:PRE-084 crystal structure reported here.

Reviewer #4 (Remarks to the Author):

In the manuscript two X-ray crystal structures of the μ 1 receptor of *X. laevis* are reported. The first structure was obtained in the presence of the typical agonist PRE-084 (xS1RPRE-084) and the second structure was obtained without ligand (xS1Rno-adding). Unfortunately, the xS1Rno-adding structure has something in the binding site, which could not be identified due to the low resolution (3.56 Å). Therefore, it is recommended to identify this bound molecule to evaluate correctly the binding status of xS1Rno-adding.

Two major differences between the structures of the μ 1 receptor with agonist (xS1RPRE-084) and without ligand (xS1Rno-adding) were detected: the orientation of the Tyr203 side chain and the positioning of the α 4 and α 5 helices towards each other. This observation might be important to understand the pathway of ligands entering the binding pocket.

The interaction of the amino acid residues of Tyr203 (α5 helix) and Leu179 (α4 helix) was analyzed by mutation of these amino acids into Cys residues respectively. The corresponding Cys residues were then coupled with maleimide conjugated with a 5 kDa PEG unit. The authors should commend, why they chose this modification. It is expected that the introduction of such a big modification at a crucial position within the α1 receptor might totally destroy the structure and properties. The very small differences recorded by ITC are very surprising. Moreover, the Western Blot in Figure 4a shows that only a small fraction of the protein has been modified. The results of this experiment needs further explanation and discussion before publication.

The results describing the structural differences between xIS1RPRE-084 and xIS1Rno-adding are very interesting and will stimulate the further α1 receptor research. Nevertheless, the proof of the gating properties of Tyr203 (α5 helix) and Leu179 (α4 helix) need further explanation and discussion, which is required before publication in "Nature Communications".

Minor comments

1. "Carboxy" group instead of "carboxyl" (remove the l at the end.).
2. Page 3, line 45: "that its opening requires.." instead of "open".
3. Figure 3e: the colors light green and light blue are very similar. It is suggested to use colors, which could be easier differentiated.
4. Page 5, line 97: "allowing an opening between helices.." without "to form".
5. Page 5, lines 111,112 and line 117: The number of binding sties should be given more precisely. It is assumed that the given number means binding sites per single protein?
6. Page 5, lines 118, 121: The valid digits of Kd values (e.g. $1.024 \pm 0.050 \mu\text{M}$ to $1.950 \pm 0.045 \mu\text{M}$) and number of binding sites should be checked very carefully (too many digits).
7. The correct SI units should be used, e.g. h instead of hour, min instead of minutes, s instead of seconds, etc.
8. In the legend of Figure 4, Western blot should be mentioned for Figure 4a.
9. Line 166: The correct IUPAC name of PRE-084 is: [2-(morpholin-4-yl)ethyl] 1-phenylcyclohexane-1-carboxylate.
10. The abbreviations in Figure 2 should be defined in the legend: X. laevis, D. rerio, G. gallus, M. musculus, H. sapiens.

Reviewer #1:

1. The manuscript by Meng et al titled “An open conformation of sigma-1 receptor” reports crystal structures of xenopus sigma-1 receptor in complex with either a known agonist PRE-84 or an endogenous unidentified ligand. The first structure is similar to the previously reported structure of human S1R save for small changes in the TM helix, but the second structure shows a different conformation of the two C-terminal helices so that the access to the ligand binding pocket becomes obvious. The authors then modified the residues on the C-terminal helices and showed that agonist binding is compromised. Because the C-terminal helices were thought to form a tight bundle through the “hydrophobic zipper”, previous studies have assumed that the two would stay together during the agonist binding process. The new structure provides a fresh perspective on what structural changes are possible and thus is highly significant. The study provides a simple interpretation of ligand access to its binding site on S1R, and because S1R is involved in multiple diseases and is a drug target, these results will have a significant impact.

The diffraction data and the refinement of structural models are of good quality with statistics expected for the modest resolution of the data. It would be necessary to show the regions of the electron density map corresponding to the beta roll and the helices in supplementary figures to provide confidence on structures.

Response:

Thanks for the comment and suggestion. Seven crystal structures are now reported in the revised manuscript, including the two structures submitted originally. We have added electron density maps to show regions of the β barrel and the helices for both the closed (Supplementary Fig. 2b, 2d) and the open-like (Fig. 3c, 3e and Supplementary Fig. 3b) conformations.

2. I appreciate authors effort to validate the conclusions from the structure with functional analyses. However, I am concerned that modification by the mPEG-Mal-5k has low efficiency and that there are higher molecular weight bands on the gel. I suggest that the authors consider these points, although #3 and #4 are not meant as a demand for additional experiments.

(1) Is it possible that the endogenous ligand prevents the modification? If so, perhaps extensive dialysis would empty the binding pocket and enhance the modification.

Response:

Thanks for the comment and suggestion. We apologize for not making this clear in the original submission. We were aware of the relatively low modification efficiency of mPEG-Mal-5K to Cys179 or Cys203 in α 1R, and we have attempted to test if binding of the endogenous ligand

would affect the modification efficiency (Response Fig. 1). However, extensive dialysis of the $\text{x}\sigma\text{1R-C179}$ or $\text{x}\sigma\text{1R-C203}$ sample did not enhance mPEG-Mal-5K modification (compare lane 1 to lane 5 for C179, and lane 3 to lane 7 for C203). Furthermore, addition of the ligand PRE084 to the dialyzed $\text{x}\sigma\text{1R-C179}$ or $\text{x}\sigma\text{1R-C203}$ sample produced no appreciable effect on modification efficiency (compare lane 1 to lane 2 for C179, and lane 3 to lane 4 for C203). Similarly, adding PRE084 to the undialyzed $\text{x}\sigma\text{1R-C179}$ or $\text{x}\sigma\text{1R-C203}$ sample had no apparent effect on modification (compare lane 5 to lane 6 for C179, and lane 7 to lane 8 for C203). Therefore, it seems less likely that the ligand binding hindered mPEG-Mal-5K modification. Meanwhile, on the contrary, we noticed that the dialysis reduced modification (compare lane 1 to lane 5 for C179, and lane 3 to lane 7 for C203). We suspect that the dialysis emptied the ligand binding pocket of $\text{x}\sigma\text{1R}$ and caused the receptor to adopt a more occluded conformation, which is a following project currently under investigation in the lab.

On the other hand, the exterior face of the carboxy-terminal two-helix bundle of σ1R is hydrophobic and membrane-adjacent. Therefore, detergents are expected to cover this region when σ1R is purified. In our opinion, these nearby detergents would impede the mPEG-Mal-5K access to Cys179 or Cys203, which resulted in limited modification efficiency. We have added this description in the revised text.

Response Figure 1. Modification of $\text{x}\sigma\text{1R-C179}$ or $\text{x}\sigma\text{1R-C203}$ in the indicated conditions by mPEG-Mal-5K. The wild-type (WT) $\text{x}\sigma\text{1R}$ served as the control. The pound sign (#) indicates that the protein sample contains the C91S mutation and has no native cysteine. Dialysis was repeated twice in 1 l buffer for 24 h at 4 °C. Some protein samples containing 1 mM PRE084

before mPEG-Mal-5K treatment were indicated above the gel. Protein samples were analyzed by SDS-PAGE and Coomassie blue staining. The black arrowheads indicate the band positions without and with one mPEG-Mal-5K modification. The red arrowhead indicates a minor impurity band near 50 kDa, which does not affect the gel analysis.

(2) The authors should clearly state what is the expected behavior for an xl σ 1R modified with mPEG-Mal-5k at positions 179, 182, or 203. Does the loss of binding site N correlate with the percentile of modification? And finally, why is there a two fold change in K_d after the modification if only a small fraction of S1R is modified?

Response:

Thanks for the suggestion. In an ideal situation, the opening between α 4 and α 5 of the modified xl σ 1R-C179 or xl σ 1R-C203 protomers would be blocked by the bulky modifier (mPEG-Mal-5K), while the other, unmodified protomers would bind ligands like wild-type xl σ 1R. Therefore, the number of available binding sites per protomer (stoichiometry) measured by ITC is expected to decrease in the modified samples, and this reduction is expected to correlate with the modification percentage. This description has been added in the revised text as suggested.

In reality, we have observed an ~ 44% reduction in the modified xl σ 1R-C179 sample and an ~ 21% reduction in the modified xl σ 1R-C203 sample for the stoichiometry (Fig. 4e, 4f). The mPEG-Mal-5K modification was estimated to be ~ 23% for xl σ 1R-C179 and ~ 13% for xl σ 1R-C203 (Supplementary Fig. 4e), which correlates with the reduction of the available binding site number but the values are smaller. Meanwhile, due to the extended shape of mPEG-Mal-5K, it is possible that one mPEG-Mal-5K molecule may block more than one xl σ 1R protomer, which may explain why the reduction percentage is higher than the modification percentage. This description has been added in the revised text.

Previous studies have suggested a cooperative ligand binding mode in σ 1R (see ref. 20). Consistently, we observed an elevated equilibrium dissociation constant (K_d) in the modified xl σ 1R samples, indicating that the modified xl σ 1R protomers impaired ligand binding in the other, unmodified protomers. Meanwhile, σ 1R has been shown to exist as multimers in solution (e.g. trimers as in the structure). Therefore, although only a relatively small fraction of xl σ 1R protomers were modified, it is possible that all the multimers that contained even one modified protomer would be impaired for ligand binding, thus affecting a much larger fraction. We have added this description in the revised text.

(3) Could the authors find or synthesize a derivative of a known agonist that would extend between α 4 and α 5? This would strengthen the conclusion from the structure.

Response:

Thanks for this great suggestion. We found such a ligand with extended length (DIM-3C) that may fulfill this purpose (see ref. 21). Dr. Collina and Dr. Rossino at University of Pavia kindly provided us ~ 2 mg of DIM-3C for soaking the $\text{xI}\sigma_1\text{R}$ crystals. However, DIM-3C is highly hydrophobic and could only be dissolved with organic solvents such as methanol and dimethyl sulfoxide, which damaged the $\text{xI}\sigma_1\text{R}$ crystals. Unfortunately, no structure was obtained for the DIM-3C soaking experiment. In the meantime, we searched the online inventory for $\sigma_1\text{R}$ antagonists and found S1RA (E-52862) hydrochloride with a relatively extended shape. In the end, S1RA was successfully soaked in two $\text{xI}\sigma_1\text{R}$ structures. One is $\text{xI}\sigma_1\text{R}_{\text{closed-S1RA}}$ in the closed conformation (Supplementary Fig. 2e). The other is $\text{xI}\sigma_1\text{R}_{\text{C179/C203-S1RA}}$, in which one protomer contained a possible S1RA molecule between α_4 and α_5 (Supplementary Fig. 5b). Both new structures have been reported in the revised manuscript and this description has been added in the Discussion section.

(4) Given that single cysteine mutations are functional, would it make sense to make a double cysteine mutation, for example, C179/C203, and test whether ligand binding is affected in the presence and absence of cys bridge? Or alternatively, whether the formation of a cys bridge is affected by the presence of a ligand?

Response:

Thanks for this great suggestion. We generated the $\text{xI}\sigma_1\text{R-C179/C203}$ mutant, and performed both functional and structural studies as suggested. The key functional results are shown in Fig. 4b, 4c and 4d. Briefly, the disulfide bond formation significantly reduced the number of available binding sites per protomer (stoichiometry), and the stoichiometry could be partially reverted when the oxidized sample was treated with β -mercaptoethanol. Furthermore, the binding site number correlates with mPEG-Mal-5K modification that assesses disulfide formation in the protein samples. This data has been described in detail in the Results section, and is the major functional validation in the revised manuscript for this study.

3. Overall, the manuscript is clear and has a good logical thread. However, there are multiple grammatical errors that I do not expect to find in a manuscript with an inspiration for Nature Communications. Professional editing or enlisting help from a native English speaker may solve the problem.

Response:

Thanks for the suggestion. We have invited a colleague, Dr. Quick at Columbia University, for critical reading of the manuscript and have corrected multiple grammatical errors as suggested. We will further consider a professional editing service if necessary.

4. Finally, I would like to suggest that the authors consider changing S1R_{no-adding} to S1R_{endo} for S1R with an endogenous ligand.

Response:

Thanks for the suggestion. Seven structures are now reported in the revised manuscript, and both the original structures have been renamed. xIS1R_{PRE-084} has been changed to xIσ1Rclosed-
PRE084, and xIS1R_{no-adding} has been changed to xIσ1R_{open-endo} as suggested.

Reviewer #2:

1. Meng et al. reports crystal structures of sigma1 receptor with PRE084 and unknown ligand carried over from purification step. Given previously reported crystal structure of human S1R displays a closed conformation in which ligand is occluded in the cupin-fold, it has been questioned how ligand enters to its binding site. The authors claim that previously proposed open mechanism from MD simulation is unlikely to occur, and analyzed 3.56Å resolution crystal structure in an open state but bound unknown substrate in it. In this structure, Y203 side chain points to the intrinsic ligand, and as a consequence, binding pocket becomes solvent-accessible.

It is therefore unclear whether the proposed “gate (Y203-L179)” is a canonical ligand entry site or just induced by the binding of unknown ligand. However, they fall short seriously in their presentation, and the functional evidences are not conclusive to show L179-Y203 gating.

Response:

We thank the reviewer for raising this important question, as it motivated us to provide more experimental evidence to strengthen the manuscript. In this revised manuscript, seven structures (including the original two) for $\alpha 1$ R were reported, including three structures in the closed conformation, three in an open-like conformation, and one for the C179/C203 double mutant. Functional analysis of the opening between $\alpha 4$ and $\alpha 5$ was carried out in two parts: one is to prevent the opening from formation by disulfide crosslinking; the other is to block the opening as in the original submission. The structural and functional data are consistent in this study and support the opening between $\alpha 4$ and $\alpha 5$ as the ligand entry site in $\sigma 1$ R.

The revised manuscript has been expanded to a full article format, including now 5 figures, 1 table, and 8 supplementary figures for a better presentation. In the revised manuscript, the original open structure ($\alpha 1$ S1R_{no-adding} in the original submission) was renamed to $\alpha 1$ R_{open-endo}, and two additional open structures bound with the ligand PRE084 ($\alpha 1$ R_{open-PRE084-co} and $\alpha 1$ R_{open-PRE084-soak}) were reported. The three open-like structures are nearly identical, suggesting that the opening between $\alpha 4$ and $\alpha 5$ is less likely an artifact induced by an unknown molecule. During the revision, we also recognized our inappropriate usage of the word ‘gate/gating’ in the original submission. The word ‘gating’ usually refers to the regulation of the ligand by the entrance, which was not what our data intended to support, as our main claim of the original submission was to define the ligand entry site in $\sigma 1$ R. Therefore, in the revised manuscript, we removed ‘gate/gating’-related language and focused on the ligand entry pathway between $\alpha 4$ and $\alpha 5$ in $\sigma 1$ R.

Major points

2. It is unclear how three monomers in the trimer are similar or identical to each other. According to the crystal symmetry, each three monomers are not necessarily identical. If these are identical, authors could apply NCS to improve the density map. The process was not described in either Methods or main text. If monomers show different conformation, structures of three monomers must be compared, especially around gate structure.

Response:

Thanks for the suggestion. We apologize for not making this clear in the original submission. Due to the modest resolutions, both structures in the original submission were refined with NCS applied throughout the refinement. In the revised manuscript, seven structures are now reported. NCS was also applied throughout the refinement due to their limited resolutions, except that for $xl\sigma_1R_{open-PRE084-co}$ (3.10 Å) and $xl\sigma_1R_{open-PRE084-soak}$ (2.85 Å), the NCS restraints were relaxed in the last a few rounds of refinement. Conformational difference between different protomers of $xl\sigma_1R_{open-PRE084-co}$ and $xl\sigma_1R_{open-PRE084-soak}$ were described, especially the rotamer dynamics of Tyr203 in the same structure (Fig. 3e, 3f). This description has been added in the Methods section and the Results section as suggested.

3. In the relatively low-resolution map, side chain rotamers are not unequivocally determined. Author should present electron density maps for not only particular residues, but also whole molecule especially around gate position to show how Y203 rotamer is defined in the structure.

Response:

Thanks for the suggestion. Seven structures are now reported in the revised manuscript, and we have added electron density maps for both the closed (now Supplementary Fig. 2b, 2d) and the open-like (Fig. 3c, 3e and Supplementary Fig. 3b) structures, as well as the $\alpha 4/\alpha 5$ region (Fig. 3c, 3e, 3f), as suggested.

4. Figures are not well organized and drawn. Color codes are confusing. For example, in Fig1d, alpha4 helix and some beta-sheets are drawn in orange, but this is not indicated in the figure caption. In Fig. 1a left, author used green and magenta for the position to disrupt in the previous simulation, and similar green and pink colors are used for the to the right panel.

Response:

Thanks for this comment. We apologize for not making the figures better presented in the original submission. In the revised manuscript, we have redrawn all figures and carefully chose each color for a clearer presentation as suggested.

5. In the present crystal structure, the $\alpha 1$ helix is significantly tilted along with the membrane plane, which authors mentioned artifact due to crystal packing. Compared with previous crystal structure in which $\alpha 1$ helix locate enough far from $\alpha 4-5$ with 2 or 3 OLC molecule in between, present $\alpha 1$ seems closer to $\alpha 4-5$. This is not clear from the figure. Does $\alpha 1$ helix asserts any effect on the binding pocket especially via $\alpha 4-5$ helices?

Response:

Thanks for this question. We have redrawn the figures to compare $\alpha 1$ of the closed and the open-like conformations of $x\sigma 1R$, and $\alpha 1$ of the human $\sigma 1R$ structure (Fig. 3a, 3b). The new figures now show more clearly the position of $\alpha 1$ relative to the rest of the receptor. Indeed, the $x\sigma 1R$ $\alpha 1$ is closer to the β barrel and the ligand binding site than in the $h\sigma 1R$ structure. In all $\sigma 1R$ structures, the $\alpha 1$ helix does not interact directly with $\alpha 4$ or $\alpha 5$. However, in the closed conformation of $x\sigma 1R$, Trp26 and Leu27 of $\alpha 1$ form hydrophobic interactions with a Leu97-containing loop from the β barrel (Supplementary Fig. 8a, 8b). We discussed this feature in detail in the Discussion section in the revised manuscript.

6. It is unclear that the binding site in the S1Rno-adding is fully open to the solvent. It seems that the gate (L179-Y203) is facing to the membrane surface. Furthermore, even if Y203 took different rotamer conformation, the loop structure between beta-sheet (S99,L100,S101 in human receptor 5hk1, see figure) blocks ligand dissociation. In the figures author provide (especially Fig3d) are only showing part of pocket, and thus unfair and misleading.

*Attached Figure human S1R (pdb:5hk1)

L182 and Y206: magenta, S99, L100 and S101: Yellow (pdb 5hk1)

Response:

Thanks for this great comment. Indeed, the hydrophobic exterior face of the $\alpha 4/\alpha 5$ helices indicates that $\alpha 4/\alpha 5$ are likely membrane-adjacent, and therefore, the ligand binding site of an open $\sigma 1R$ is probably not fully open to the solvent within cells. We apologize for having caused confusion by using the term 'solvent access' in the original submission, as we intended to describe a pathway connecting the ligand binding site and the outside milieu of the receptor. In the revised manuscript, we changed the text of 'solvent access' to 'ligand entry pathway'.

Meanwhile, the ligand entry pathway between $\alpha 4$ and $\alpha 5$ (PATH2 in the revised manuscript) suggests that the ligand would access to, and dissociate from, the ligand binding site through

the membrane. Given that the σ 1R ligands are largely hydrophobic, and previous studies show that the synthetic σ 1R ligands need to partition in and pass through cell membranes to reach σ 1R (see ref. 22), we cautiously discussed this aspect in detail in the Discussion section.

Leu97 in xl σ 1R (corresponding to Leu100 in h σ 1R) is one of the residues surrounding the opening between α 4 and α 5 (Fig. 4a). We also apologize for not making the Leu97-containing loop clearer in the original submission and have updated related figures (Fig. 3d, 4a and Supplementary Fig. 8b) in the revised manuscript as suggested. The Leu97-containing loop in xl σ 1R forms hydrophobic interactions with α 1 in the closed conformation, and this interaction is lost in the open-like structures (Supplementary Fig. 8a, 8b). Based on this structural change, we cautiously discussed a possibility that conformational flexibility of the Leu97-containing loop may facilitate ligand access to, and dissociation from, the ligand binding site from within the membrane (see Discussion for details).

7. It is unclear why the binding capacity of human receptor is around 0.5 in contrast to xenopus receptor (~1.0).

Response:

Thanks for this question. During expression and purification of xl σ 1R and h σ 1R, we noticed better protein stability for xl σ 1R than h σ 1R in our experiments. For example, a fraction of h σ 1R would stick on the cobalt beads during the affinity purification step and could not be eluted, presumably due to denaturation issues. A fraction of purified h σ 1R would precipitate in sizing columns if stored at 4 °C overnight, possibly due to stability issues. Meanwhile, purified xl σ 1R is stable for days when stored at 4 °C. We suspect that a fraction of the purified h σ 1R is not competent for ligand binding, which may explain the lower stoichiometry in ITC compared to xl σ 1R. Therefore, we focused on presenting only the xl σ 1R data in the revised manuscript to make it more straightforward, more concise and clearer.

8. Label ratio of used sample must be determined and compared to the binding experiment. Judging from SDS-PAGE, label ratio of human receptor seems less than 20% but the effect of ITC analysis seems too large.

Response:

Thanks for this comment and suggestion. The modification percentage of xl σ 1R cysteine mutants by mPEG-Mal-5K has been determined by band intensity analysis (see Methods for details), and compared to the ITC data as suggested. In the modification experiment, we have observed an ~ 44% reduction in the modified xl σ 1R-C179 sample and an ~ 21% reduction in the modified xl σ 1R-C203 sample for the stoichiometry (Fig. 4e, 4f). The mPEG-Mal-5K modification

was estimated to be ~ 23% for xl σ 1R-C179 and ~ 13% for xl σ 1R-C203 (Supplementary Fig. 4e), which correlates with the reduction of the available binding site number but the values are smaller. Meanwhile, due to the extended shape of mPEG-Mal-5K, it is possible that one mPEG-Mal-5K molecule may block more than one xl σ 1R protomer, which may explain why the reduction percentage is higher than the modification percentage. This description has been added in the revised text.

In the meantime, we were aware of the limited modification efficiency by mPEG-Mal-5K as a means to block the opening between α 4 and α 5 of xl σ 1R. Therefore, we designed another experiment to form a disulfide bridge between C179 and C203 to test the opening. The key functional results are shown in Fig. 4b, 4c and 4d. Briefly, the disulfide bond formation significantly reduced the number of available binding sites per protomer (stoichiometry), and the stoichiometry could be partially reverted when the oxidized sample was treated with β -mercaptoethanol. Furthermore, the binding site number correlates with mPEG-Mal-5K modification that assesses disulfide formation in the protein samples. This data has been described in detail in the Results section, and is the major functional validation in the revised manuscript for this study.

9. Why affinity changes upon labeling? If L179 or Y203 act as a gate, modification with such a huge reagent may lost its binding ability, which reflect reduced number of binding site (Fig4f). In other words, unlabeled receptor must be healthy and accordingly affinity of receptor that accept ligand must be the same before and after modification. Is it an allosteric effect?

Response:

Thanks for this question and comment. Indeed, the equilibrium dissociation constant (K_d) is not expected to change if the individual xl σ 1R protomer binds its ligand independently. Previous studies have suggested a cooperative ligand binding mode in σ 1R (see ref. 20). Similarly, we observed an elevated equilibrium dissociation constant (K_d) in the modified xl σ 1R samples, indicating that the modified xl σ 1R protomers impaired ligand binding in the other, unmodified protomers. Our data is consistent with the cooperative ligand binding mode proposed in previous studies (see ref. 20). We have added this description in the revised text.

10. From the kinetic analysis in ref 9, ligand-independent conformational change is proposed. If L179-Y203 is a gate, their simple mutation (alanine or glycine) may affect ligand binding kinetics. Related to this comment, does Cys replacement of highly conserved L179 and Y203 exert any difference in ligand binding or its kinetics?

Response:

Thanks for this comment and question. The ligand-independent conformational change of σ 1R suggests a preparative step of the receptor from a 'ground', ligand-inaccessible state to a ready-for-binding state during ligand binding (now ref. 20). In a serendipitous experiment, we found that dialysis of purified xl σ 1R resulted in the receptor being more resistant to chymotrypsin digestion (compare lane 4 to lane 3, Response Fig. 2). This data suggests that the dialyzed xl σ 1R adopts a more compact conformation than the available σ 1R structures. Since an unknown molecule has been observed to bind in purified xl σ 1R, we postulated that the dialysis may empty the ligand binding pocket of xl σ 1R and the receptor may have entered a 'ground' state. Moreover, addition of the ligand PRE084 had no appreciable effect in chymotrypsin treatment of the dialyzed protein (compare lane 6 to lane 4, Response Fig. 2), suggesting that the conformational change from the dialyzed, 'ground' state to the before-dialyzed state is ligand-independent. Whether this dialyzed, 'ground' state of xl σ 1R represents the ligand-inaccessible state in the ligand-independent conformational change proposed previously (see ref. 20) is a following project currently under investigation in the lab.

Meanwhile, the single cysteine mutations (Cys179 or Cys203) or the double cysteine mutation (Cys179/Cys203) did not change significantly the ligand binding characteristics in xl σ 1R such as stoichiometry and affinity (Fig. 4b, 4c, 4e, 4f), which are the most pertinent and direct parameters to assess the ligand entry pathway in σ 1R. Since the ITC data have fulfilled our main purpose in this study, we did not perform further kinetic analysis, which may be carried out as we investigate the dialyzed, 'ground' state in the future.

Response Figure 2. Resistance of wild-type $\alpha 1R$ to chymotrypsin in the indicated conditions. Dialysis was repeated twice in 1 l buffer for 24 h at 4 °C. The protein samples (lane 5 and 6) contained 1 mM PRE084 before chymotrypsin treatment were indicated above the gel. Protein samples were treated with 1/10 α -chymotrypsin (w/w) for 10 min at 20 °C and the reaction was stopped by addition of PMSF to a final concentration of 1 mM. Protein samples were analyzed by SDS-PAGE and Coomassie blue staining. The black arrowheads indicate the band positions of undigested (Full-length) and digested (Fragment) $\alpha 1R$ protein.

Reviewer #3:

1. The manuscript presented by Meng et al. describes two crystal structures of S1R from *Xenopus laevis* (xlS1R). The first structure was obtained in the presence of the agonist PRE-084, exhibiting a conformational state consistent with a previously reported human S1R structure bound to (+)-pentazocine. The main difference is observed into the transmembrane helix alpha1, which is attributed to its flexibility or crystal packing. The second structure was crystallized in the absence of any known S1R ligand; however, the authors found an “unknown molecule” presumably from an endogenous origin located at the ligand-binding pocket. A structural comparison between the two xlS1R crystallized structures revealed a conformational change in two alpha-helices at the C-terminus (alpha4-alpha5), which is attributed to an apparent opening of the ligand-binding pocket towards the ER membrane. Two residues were identified as key in the opening process (i.e. L179 and Y203), which were mutated by cysteine affecting the ligand-binding properties in S1R.

This work could represent a step forward understanding the conformational changes associated with the ligand binding in the Sigma 1 receptor expressed in different organisms; however, not knowing the chemical nature of the molecule responsible of the conformational changes in xlS1R_{no-adding} makes this work unconvincing. The major concern is about the conclusion that an “open structure” of S1R has been found. The resolution of the crystal structure and the nonspecificity of the bound ligand makes not clear that the structure reported here is in open state. Further studies are required to identify the molecule bound to the ligand-binding pocket in this conformation and to demonstrate that this binding event is not a crystallization artifact.

Response:

We thank the reviewer for this great comment and suggestion. We have performed extensive, additional experiments to strengthen this manuscript. The revised manuscript has been expanded to a full article format, including now 5 figures, 1 table, and 8 supplementary figures. In the revised manuscript, the original open structure (xlS1R_{no-adding} in the original submission) was renamed to xlσ1R_{open-endo}, and two additional open structures bound with the ligand PRE084 (xlσ1R_{open-PRE084-co} and xlσ1R_{open-PRE084-soak}) were reported. The three open-like structures are nearly identical, suggesting that the opening between α4 and α5 is less likely an artifact induced by an unknown molecule. Meanwhile, we also changed the word ‘open’ to ‘open-like’ for a more cautious term to describe a conformational state of σ1R that forms an opening between α4 and α5, since it is possible that the reported open-like structure in this study may not represent the fully-open state.

2. Authors stated that the current resolution of xIS1Rno-adding (which is very low 3.56 Å) limited their ability to identify the bound molecule. However, they consider that this resolution was enough to identify the residues Tyr203 and Leu179 shifted generating an open conformation.

Response:

Thanks for this comment. We apologize for not making this clear in the original submission. It is usually difficult to reveal the molecular identity of an unknown molecule based on solely the shape of an electron density, especially if the structure resolution is modest. However, it is fairly common to unequivocally place the side chains in a modest-resolution density map, as long as the region of interest is well resolved. For example, even a 3.8 Å-map could have many regions well packed and resolved, while usually the flexible loops or ends will have poor details. In the xI σ 1R structures, the α 4/ α 5 region is well resolved. Furthermore, in the revised manuscript, we reported two additional open-like structures complexed with the ligand PRE084 at higher resolutions (3.10 Å and 2.85 Å), and the α 4/ α 5 residues were more confidently placed.

3. At first glance, the xIS1Rno-adding structure does not show a significant opening of the alpha4-alpha5 helices to accommodate ligands entering. A structural analysis exhibiting the dimensions of this cavity in the occluded and open structure would be necessary to rationalize the passing of known S1R ligands. Due to the remarkable plasticity in ligand recognition of S1R, ligands from different size and shape should penetrate the gate formed by the Leu179-Tyr203 pair.

Response:

Thanks for this suggestion. We have performed a tunnel analysis in both the closed and the open-like structures of xI σ 1R as suggested (Response Fig. 3a). The closed conformation is occluded to the outside solvent as its opening near Tyr203 is smaller than the size of water (Response Fig. 3b), while the open-like conformation has an opening of ~ 6 Å in diameter between α 4 and α 5 (Response Fig. 3c). However, this tunnel analysis is not accurate for σ 1R, as it underestimates the dimension of tunnels whose transverse section is elongated such as the one in the open-like xI σ 1R structure (Fig. 3d). We measured both the short and the long axes of the opening between α 4 and α 5 in the open-like conformation of xI σ 1R, and it spans an area of more than 10.3 Å x 5.5 Å (Fig. 3d). This opening is sufficiently large for the σ 1R ligands such as PRE084 (~ 7.5 Å x 5.5 Å measured transversely) and S1RA (~ 7.6 Å x 5.2 Å measured transversely) to pass through. This structural analysis has been added in the revised manuscript without the tunnel plot, which is not accurate for the open-like xI σ 1R structure.

Response Figure 3. Tunnel analysis for the $x1\sigma1R$ structures by the MOLEonline 2.0

webserver (<https://mole.upol.cz>). (a) An illustration of the tunnel detection starting from Glu169 (colored in magenta) to the $\alpha4/\alpha5$ region of $x1\sigma1R$ (in gray surface representation), and the blue arrow indicates the tunnel direction. (b)-(c) The tunnel radius plot along the tunnel path for the closed $x1\sigma1R$ structure ($x1\sigma1R_{\text{closed-PRE084}}$, panel b) and the open-like $x1\sigma1R$ structure ($x1\sigma1R_{\text{open-PRE084-co}}$, panel c). The black arrow indicates the approximate position of Tyr203.

4. To evaluate the importance of the residues Leu179 and Tyr203, cysteine mutations were performed, and ligand-binding properties were evaluated by isothermal titration calorimetry (ITC). Although some effects were observed, it is questionable that these assays are enough to demonstrate that modifying these positions “blocks or impedes ligand entering through the opening”. L179 and Tyr203 have shown to interact with ligands; therefore, it could be trivial to find an effect when they are replaced by Cys modified with a bulky reagent. In this regard, ITC does not demonstrate that “a blockade near residues 179/203 hinders ligand access and binding to $xS1R$ ” by means of a mechanism where Leu179-Tyr203 act as a gate. In addition, authors said: “Although wild-type $xS1R$ contains a native cysteine at residue 91 (Cys91), modification of mPEG-Mal-5K on Cys91 showed little influence on binding of the agonist PRE-084 to the wild-type $xS1R$ sample and yielded comparable active binding site numbers”. It could be because the side chain of C91 is not oriented to the ligand-binding site.

Response:

We thank the reviewer for this critical comment. We were aware of the limitations associated with mPEG-Mal-5K modification as a means to block the opening between $\alpha4$ and $\alpha5$ of $x1\sigma1R$. Therefore, we designed another experiment to form a disulfide bridge between C179 and C203 to test the opening. The key functional results are shown in Fig. 4b, 4c and 4d. Briefly, the disulfide bond formation significantly reduced the number of available binding sites per protomer (stoichiometry), and the stoichiometry could be partially reverted when the oxidized sample was treated with β -mercaptoethanol. Furthermore, the binding site number correlates with mPEG-Mal-5K modification that assesses disulfide formation in the protein samples. This data is

consistent with the modification experiment and the new structural data. This result has been described in detail in the Results section, and is the major functional validation in the revised manuscript for this study.

Minor points:

1. Authors point out that “a computational simulation further suggests that the opening of S1R occurs in a two-step conformational change that requires disruption of the cupin-fold.” It should be noted that recently another computational study (J. Chem. Inf. Model. 2020, 60, 756–765) explored two ligand pathways to enter and exit the binding site. According to this study, the most favored pathway implies deformation of the cupin-fold and the less favored pathway involves a gap between the two membrane-adjacent helices D and E (α 4 and α 5). This last pathway has common characteristics with the “open structure” reported in this manuscript. Authors should discuss about this antecedent.

Response:

Thanks for this comment and suggestion. We apologize for not having discussed this relevant study in the original submission, and have included the reference (now ref. 21) and discussed this study in the revised manuscript as suggested.

2. On the other hand, Yano et al (Neuropharmacology 2018, 133, 264-275) reported molecular modeling of several ligands interacting with S1R, including PRE-084. In this work, PRE-084 showed a unique kinetics for entering the ligand binding site. Authors should discuss whether structural characteristics are comparable with the xS1R:PRE-084 crystal structure reported here.

Response:

Thanks for this comment and suggestion. We apologize for not having discussed this relevant study in the original submission. In the revised manuscript, three PRE084-containing structures were reported: one in the closed conformation, and the other two in the open-like conformation. We discussed a possible entering/exiting process for PRE084, in which the phenyl ring of PRE084 pressures against the α 4 helix. This structural feature has also been observed in the simulations reported by Yano et al. (now ref. 22). We have added this discussion in the Discussion section as suggested.

Reviewer #4:

1. In the manuscript two X-ray crystal structures of the σ_1 receptor of *X. laevis* are reported. The first structure was obtained in the presence of the typical agonist PRE-084 (xIS1R_{PRE-084}) and the second structure was obtained without ligand (xIS1R_{no-adding}).

Unfortunately, the xIS1R_{no-adding} structure has something in the binding site, which could not be identified due to the low resolution (3.56 Å). Therefore, it is recommended to identify this bound molecule to evaluate correctly the binding status of xIS1R_{no-adding}.

Response:

We thank the reviewer for this suggestion. We have performed extensive, additional experiments to strengthen this manuscript. In the revised manuscript, the original open structure (xIS1R_{no-adding} in the original submission) was renamed to xIS1R_{open-endo}, and two additional open structures bound with the ligand PRE084 (xIS1R_{open-PRE084-co} and xIS1R_{open-PRE084-soak}) were reported. The three open-like structures are nearly identical, suggesting that the opening between α_4 and α_5 is less likely an artifact induced by an unknown molecule.

2. Two major differences between the structures of the σ_1 receptor with agonist (xIS1R_{PRE-084}) and without ligand (xIS1R_{no-adding}) were detected: the orientation of the Tyr203 side chain and the positioning of the α_4 and α_5 helices towards each other. This observation might be important to understand the pathway of ligands entering the binding pocket.

The interaction of the amino acid residues of Tyr203 (α_5 helix) and Leu179 (α_4 helix) was analyzed by mutation of these amino acids into Cys residues respectively. The corresponding Cys residues were then coupled with maleimide conjugated with a 5 kDa PEG unit. The authors should comment, why they chose this modification. It is expected that the introduction of such a big modification at a crucial position within the σ_1 receptor might totally destroy the structure and properties. The very small differences recorded by ITC are very surprising.

Response:

Thanks for this comment. In the original submission, we chose mPEG-Mal-5K as the modifier to block the opening between α_4 and α_5 mainly due to two reasons: first, it is bulky and should be able to block the opening; second, it has the advantage to allow quick assessment of the relative modification levels for the ITC samples, so that we could quickly correlate the modification with the binding data.

In the meantime, we were aware of the limitations associated with mPEG-Mal-5K modification as a means to block the opening between $\alpha 4$ and $\alpha 5$ of $\sigma 1R$. Therefore, we designed another experiment to form a disulfide bridge between C179 and C203 to test the opening. The key functional results are shown in Fig. 4b, 4c and 4d. Briefly, the disulfide bond formation significantly reduced the number of available binding sites per protomer (stoichiometry), and the stoichiometry could be partially reverted when the oxidized sample was treated with β -mercaptoethanol. Furthermore, the binding site number correlates with mPEG-Mal-5K modification that assesses disulfide formation in the protein samples. This data is consistent with the modification experiment and the new structural data. This result has been described in detail in the Results section, and is the major functional validation in the revised manuscript for this study.

3. Moreover, the Western Blot in Figure 4a shows that only a small fraction of the protein has been modified. The results of this experiment needs further explanation and discussion before publication.

Response:

Thanks for this critical comment. The exterior face of the carboxy-terminal two-helix bundle ($\alpha 4/\alpha 5$) of $\sigma 1R$ is hydrophobic and membrane-adjacent. Therefore, detergents are expected to cover this region when $\sigma 1R$ is purified. In our opinion, these nearby detergents would impede the mPEG-Mal-5K access to Cys179 or Cys203, which resulted in limited modification efficiency. We have added this description in the revised text.

4. The results describing the structural differences between $\sigma 1R_{PRE-084}$ and $\sigma 1R_{no-adding}$ are very interesting and will stimulate the further $\sigma 1$ receptor research. Nevertheless, the proof of the gating properties of Tyr203 ($\alpha 5$ helix) and Leu179 ($\alpha 4$ helix) need further explanation and discussion, which is required before publication in "Nature Communications".

Response:

Thanks for this comment and suggestion. We have performed extensive, additional experiments to strengthen this manuscript. The revised manuscript has been expanded to a full article format, including now 5 figures, 1 table, and 8 supplementary figures for a better presentation. During the revision, we also recognized our inappropriate usage of the word 'gate/gating' in the original submission. The word 'gating' usually refers to the regulation of the ligand by the entrance, which was not what our data intended to support, as our main claim of the original submission was to define the ligand entry site in $\sigma 1R$. Therefore, in the revised manuscript, we removed 'gate/gating'-related language and focused on the ligand entry pathway between $\alpha 4$

and $\alpha 5$ in $\sigma 1R$.

Minor comments:

1. “Carboxy” group instead of “carboxyl” (remove the l at the end.).

Response:

Revised as suggested.

2. Page 3, line 45: “that its opening requires..” instead of “open”.

Response:

The manuscript has been expanded to a full article format and the text has been substantially revised. The related text has been revised as suggested.

3. Figure 3e: the colors light green and light blue are very similar. It is suggested to use colors, which could be easier differentiated.

Response:

Thanks for the suggestion. The manuscript has been expanded to a full article format and all figures have been redrawn. We carefully chose each color for a better presentation.

4. Page 5, line 97: “allowing an opening between helices..” without “to form”.

Response:

The manuscript has been expanded to a full article format and the text has been substantially revised. The related text has been revised as suggested.

5. Page 5, lines 111,112 and line 117: The number of binding sties should be given more precisely. It is assumed that the given number means binding sites per single protein?

Response:

Yes, it means stoichiometry (the available binding site number per protomer). The text has been updated as suggested.

6. Page 5, lines 118, 121: The valid digits of Kd values (e.g. $1.024 \pm 0.050 \mu\text{M}$ to $1.950 \pm 0.045 \mu\text{M}$) and number of binding sites should be checked very carefully (too many digits).

Response:

Thanks for the suggestion. We reduced the valid digits to the hundredth place and double-checked the values as suggested.

7. The correct SI units should be used, e.g. h instead of hour, min instead of minutes, s instead of seconds, etc.

Response:

Revised as suggested.

8. In the legend of Figure 4, Western blot should be mentioned for Figure 4a.

Response:

In all gel figures, the visualization method (Coomassie blue staining) was mentioned in the figure legend as suggested.

9. Line 166: The correct IUPAC name of PRE-084 is: [2-(morpholin-4-yl)ethyl] 1-phenylcyclohexane-1-carboxylate.

Response:

Revised as suggested.

10. The abbreviations in Figure 2 should be defined in the legend: X. laevis, D. rerio, G. gallus, M. musculus, H. sapiens.

Response:

Revised as suggested (now Supplementary Fig. 1).

REVIEWERS' COMMENTS

Reviewer #1 (Remarks to the Author):

I do not have any further comments.

Reviewer #2 (Remarks to the Author):

The manuscript is significantly improved with additional crystal structures and complementary biochemical analysis. Soaking experiment clearly showed that the PATH1 is unlikely. Authors further confirmed their hypothesis by making disulfide-formation mutant with its crystal structure, which is consistent with complementary biochemical analyses employing either disulfide or PEG modification. Finally, authors found a protomer in the double cysteine crystal bound S1RA at the gate, though limited resolution, strongly support that there is a space wide enough to accommodate ligand at the PATH2 position. Presentation of the structures in the figures are now convincing for author's conclusion. Usage of terms, including title (now changing "open-like" from previous "open") also sounds. All of my previous concerns are appropriately replied and some are now included in the manuscript. I think now the manuscript is of suitable quality and impact to the field, and therefore recommend as a candidate for the publication in Nature Communications.

Reviewer #3 (Remarks to the Author):

The authors have addressed most of the comments satisfactorily and have improved the manuscript. This work represents a valuable contribution for understanding the ligand binding mechanisms in the Sigma 1 receptor. I therefore recommend acceptance of this manuscript for publication in Nature Communications.

Reviewer #1:

I do not have any further comments.

Response:

Thanks for the comment.

Reviewer #2:

The manuscript is significantly improved with additional crystal structures and complementary biochemical analysis. Soaking experiment clearly showed that the PATH1 is unlikely. Authors further confirmed their hypothesis by making disulfide-formation mutant with its crystal structure, which is consistent with complementary biochemical analyses employing either disulfide or PEG modification. Finally, authors found a protomer in the double cysteine crystal bound S1RA at the gate, though limited resolution, strongly support that there is a space wide enough to accommodate ligand at the PATH2 position. Presentation of the structures in the figures are now convincing for author's conclusion. Usage of terms, including title (now changing "open-like" from previous "open") also sounds. All of my previous concerns are appropriately replied and some are now included in the manuscript. I think now the manuscript is of suitable quality and impact to the field, and therefore recommend as a candidate for the publication in Nature Communications.

Response:

Thanks for the comment.

Reviewer #3:

The authors have addressed most of the comments satisfactorily and have improved the manuscript. This work represents a valuable contribution for understanding the ligand binding mechanisms in the Sigma 1 receptor. I therefore recommend acceptance of this manuscript for publication in Nature Communications.

Response:

Thanks for the comment.